# Recommendations for finite element modelling of nickel-titanium stents— Verification and validation activities

**Martina Bernini**[1,2], **Rudolf Hellmuth**[2], **Craig Dunlop**[2], **William Ronan**[1], **Ted J. Vaughan**[1] *

**1** Biomechanics Research Centre (BioMEC), College of Science and Engineering, University of Galway, Galway, Ireland, **2** Vascular Flow Technologies, Dundee, United Kingdom

\* ted.vaughan@universityofgalway.ie

## Abstract

The objective of this study is to present a credibility assessment of finite element modelling of self-expanding nickel-titanium (Ni-Ti) stents through verification and validation (VV) activities, as set out in the ASME VV-40 standard. As part of the study, the role of calculation verification, model input sensitivity, and model validation is examined across three different application contexts (radial compression, stent deployment in a vessel, fatigue estimation). A commercially available self-expanding Ni-Ti stent was modelled, and calculation verification activities addressed the effects of mesh density, element integration and stable time increment on different quantities of interests, for each context of use considered. Sensitivity analysis of the geometrical and material input parameters and validation of deployment configuration with *in vitro* comparators were investigated. Results showed similar trends for global and local outputs across the contexts of use in response to the selection of discretization parameters, although with varying sensitivities. Mesh discretisation showed substantial variability for less than 4 × 4 element density across the strut cross-section in radial compression and deployment cases, while a finer grid was deemed necessary in fatigue estimation for reliable predictions of strain/stress. Element formulation also led to substantial variation depending on the chosen integration options. Furthermore, for explicit analyses, model results were highly sensitive to the chosen target time increment (e.g., mass scaling parameters), irrespective of whether quasistatic conditions were ensured (ratios of kinetic and internal energies below 5%). The higher variability was found for fatigue life simulation, with the estimation of fatigue safety factor varying up to an order of magnitude depending on the selection of discretization parameters. Model input sensitivity analysis highlighted that the predictions of outputs such as radial force and stresses showed relatively low sensitivity to Ni-Ti material parameters, which suggests that the calibration approaches used in the literature to date appear reasonable, but a higher sensitivity to stent geometry, namely strut thickness and width, was found. In contrast, the prediction of vessel diameter following deployment was least sensitive to numerical parameters, and its validation with *in vitro* comparators offered a simple and accurate (error ~ 1–2%) method when predicting diameter gain, and lumen area, provided that the material of the vessel is appropriately characterized and modelled.

**Data Availability Statement:** All relevant data are within the paper and its Supporting information files. The CAD model of the subunit was uploaded

on ZENODO repository (10.5281/zenodo.7622055).

**Funding:** TJV reports financial support was provided by EU Framework Programme for Research and Innovation Marie Sklodowska-Curie Actions. This study is part of the BioImplant ITN project, which has received funding from the European Union's Horizon 2020 research and innovation programme under grant agreement No 813869.

**Competing interests:** The authors have declared that no competing interests exist.

# 1. Introduction

Self-expanding nickel-titanium (Ni-Ti) stents have become the gold standard of treatment across a range of vascular indications [1–5], due to the shape memory and superelastic properties of this metal alloy. In the certification process for these endovascular devices, manufacturers must provide a substantial body of pre-clinical evidence that demonstrates suitable mechanical performance of the stent and delivery system, which is described in detail by the ISO 25539 standard [6]. While the vast majority of these certification activities currently involve physical testing, the Food and Drug Administration (FDA) has for many years advocated the use of computer modelling in the regulatory evaluation of devices [7]. Finite element analysis is already one of the requirements in the certification of endovascular stents for fatigue and durability [6], and general recommendations and considerations for its use in radial compression analysis are described by ASTM-F2514 [8]. A recent report by their Senior Science Council highlights the clear ambition from the FDA to increase the role of modelling and simulation in the regulatory process [7, 9]. On the other hand, there is currently no technical standard harmonised in the EU regulatory system to demonstrate the credibility of an *in silico* trial solution [10]. For many years, a key concern around the increasing use of computational modelling in the medical device sector was the lack of clear guidance on how to assess the model credibility, namely its ability to replicate the modelled reality within a predefined tolerance. In 2018, the American Society of Mechanical Engineers (ASME) released the VV-40 "Verification and Validation in Computational Modeling of Medical Devices" [11], which establishes a risk-informed credibility assessment framework based on verification, validation and uncertainty quantification activities. The ASME VV-40 standard outlines a framework that begins with the definition of the question of interest (QOI), a specific question being addressed with a computational model, and a context of use (COU), which specifies the role of modelling and simulation in addressing the QOI. It also defines a method to quantify the model risk, that is the possibility that the model may lead to a false/incorrect conclusion about device performance resulting in adverse outcomes. Verification and validation are common concepts used to describe the procedure for assessing that a product, service or system meets requirements and specifications defined for its intended purpose [12]. The verification procedure evaluates whether the product, service or system conforms to its specifications. The validation procedure ensures that the product, service or system complies with the requirements or expectations of current and potential stakeholders (e.g., customers). In the context of computational models and the ASME VV-40 standard, these concepts assume a specific meaning, whereby verification ensures a computational model is correctly implemented with respect of the conceptual model (e.g., it matches the specifications and assumptions for the given purpose of its application). On the other hand, validation assesses the accuracy of the model in representing the real system, consistently with the intended application of the model. While the ASME VV-40 document now represents one of the pillars of standards to determine model credibility in the regulatory decision-making process, there are limited examples of the effective application of the VV-40 framework to Ni-Ti-based vascular devices, despite extensive research on these implants.

The objective of this study is to present a credibility assessment of finite element modelling of self-expanding Ni-Ti stents through verification and validation activities, as set out in the ASME VV-40 standard. In this paper, we first provided a review of recent computational studies that predict the mechanical performance of self-expanding Ni-Ti stents. Based on this, we conducted a range of VV activities for several commonly investigated COUs for Ni-Ti devices. These COUs included simulated mechanical bench tests, device deployment in idealised and realistic vessels and fatigue-behaviour assessment. Within the verification activities, we identified the key model sensitivities for each COU relating to pertinent issues, such as spatial and

temporal discretization, and element formulation. Within the validation activities, we performed a sensitivity analysis to model inputs, namely geometrical and material parameters in radial compression modelling. Finally, we used *in vitro* comparators for assessing the configuration after the deployment in a vessel.

## 2. Review of computational modelling of nickel-titanium endovascular devices

Computational modelling has been used to predict a range of mechanical characteristics of endovascular Ni-Ti-based devices, from *in silico* bench testing to assess functional mechanical performance [13], predictions of fatigue behaviour [14] and deployment in patient-specific geometries [15, 16]. While the methodologies followed appear to be well-established and generally aligned with guidance in commercial software manuals [17], few studies have presented a detailed verification and validation of their modelling frameworks. **Table A1** in the S1 Appendix presents a summary of recent computational studies that have simulated the performance of Ni-Ti-based devices. It includes details of the model framework, solver settings, model discretisation, input and output data, and verification and validation activities. This summary highlights similarities and inconsistencies in the modelling approaches used, which are in some cases not even fully described, despite the published guidelines on reporting of computational models being now available [18].

Given the significant nonlinearities that are encountered due to the presence of large deformations and complex interactions, the use of explicit solver [13, 15, 16, 19–23] is preferred over implicit approaches [24–26] when modelling Ni-Ti devices. However, explicit solution schemes are conditionally stable, and many studies have applied generic criteria to ensure quasi-static conditions, which require the ratio of kinetic energy to internal energy to be kept below 5% [16, 21, 27], with additional numerical damping added if required [22]. The stable time increment in models of this type ranges from 4E-06 s to 1E-08 s [16, 28], with these values depending greatly on the boundary conditions and loading rate in use. While a minimum density of $4 \times 4$ hexahedral elements across the stent strut (e.g., width $\times$ thickness) has been suggested [17, 19, 27] with linear reduced integration [15, 16, 23, 27], the details of mesh density and element type details are sometimes not stated [20] or have been selected based on previous literature [23], with the procedure and outcomes of the verification activities rarely made available [21, 29, 30]. Also, there is substantial variation in how both geometric and material model inputs are derived. In some cases, the geometry of the device is obtained directly from the manufacturer [21, 22]. As this information can be commercially sensitive, many authors have been forced to reconstruct the geometry through imaging techniques such as optical microscopy or micro-computer tomography (micro-CT) [13–15, 19], which might add substantial uncertainty. To model the superelastic behaviour of Ni-Ti, the constitutive law from Auricchio and Taylor [31] is universally used [14–16, 19–25, 27, 28, 32–34], with this model now available as a built-in option in several commercial software packages such as Abaqus, ANSYS and COMSOL Multiphysics [35]. However, this constitutive model presents challenges as it requires up to 15 independent parameters to be determined [36]. In some studies, these have been derived from uniaxial tensile testing of dog-bone specimens [14], tube sections [20], wires/multi-wire specimens [21], or have been provided directly by the manufacturer [16, 22]. In many other studies, these parameters have been calibrated directly to the device response or have been obtained from literature [15, 19, 32, 33], which in some cases uses source data from devices with distinctly different designs and applications. Together, these verification aspects lead to several issues that create uncertainty around the model set-up that requires careful consideration and highlight the need for robust validation efforts for individual models.

Validation is generally addressed by comparing the computational model with a comparator, which could be animals, cadavers, patients, *in vitro* specimens, or phantoms. When the model is developed to guide decisions during the design phase of the device, common validation strategies make use of *in vitro* mechanical bench tests, such as radial compression, axial tension, and bending. For example, McKenna *et al.* [13, 27] developed a FE model of braided and laser-cut self-expanding Ni-Ti stents, which was compared against experiments through quantitative (e.g., radial and axial tests) and qualitative (e.g., bending configurations) data and applied the model to predict mechanical behaviour on alternative stent designs and polymer covers. However, when models are developed to address the clinical performance of a device, further validation activities are required to ensure that interactions with a vessel are correctly accounted for. With this in mind, Bernini *et al.* [37] calibrated stent material parameters through the device radial response and subsequently validated the model performance with an *in vitro* comparator of deployment in a straight silicone vessel. A similar approach was previously developed by Berti *et al.* [38], although for a FE model of a platinum-chromium stent for coronary applications. Here, the model was validated against *in vitro* experiments of deployment in idealised straight and bifurcated vessels, and in more realistic patient-specific anatomies obtained using 3D-printing technology. The authors suggested several relevant parameters to use in the comparison, which included outer diameter gain and stent malapposition, and emphasised the importance of mechanical characterisation of the vessel material to ensure that the radial response was in a physiologically relevant range. While this is an excellent example of model validation carried out for coronary devices, the application of the VV-40 standard to self-expanding Ni-Ti-based devices is limited, despite their widespread use.

In the context of VV-40, only a few studies have clearly addressed aspects of verification and validation for Ni-Ti cardiovascular devices. Luraghi *et al.* [22, 29] applied the ASME VV-40 standard to address the capability of a FE model of a stent-retriever device with the objective of predicting revascularization outcomes in realistic scenarios. Verification activities were carried out to support the choice of suitable element discretization and formulation, and validation was achieved through *in vitro* thrombectomy tests in idealized funnel- and U-shaped tubes, and more realistic patient-specific vessels. Zaccaria *et al.* [21] adopted the ASME VV-40 standard to identify the model risk and define a suitable *in vitro* experimental campaign for a Ni-Ti venous system to treat compression disorders. The authors established a framework to perform verification and validation activities applicable to a stent-like device. Verification activities quantified the effects of mesh density and element formulation on local and global parameters, namely stress and force. A comprehensive report on the method used to derive geometrical and material parameters was also included in the **Table A1** in S1 Appendix and model validation was addressed within the context of device delivery (e.g., assembly, retraction and deployment in a realistic vessel) by comparing among *in silico* and *in vitro* outcomes. Beyond these examples, the vast majority of other computational studies do not include any form of independent validation of their model prediction. Despite this, computational models of Ni-Ti stents have been widely used to predict the performance of stents in complex scenarios, including implantation behaviour in patient-specific vessels and fatigue performance. However, the predictive power of these models remains questionable without rigorous assessment of model credibility through appropriate verification and validation activities.

# 3 Materials and methods

## 3.1 ASME VV-40 standard

The ASME VV-40 standard provides a risk-informed credibility assessment framework through verification, validation, and uncertainty quantification activities for computational

**Table 1. ASME VV-40 standard process flow.**

| Process Step | Process Component | Credibility Factors |
|---|---|---|
| **Verification** | Code | Numerical code verification |
| | | Software quality assurance |
| | Calculation | Discretization error |
| | | Numerical solver error |
| | | Use error |
| **Validation** | Model | Model Form |
| | | Model Inputs |
| | Comparator | Test Samples |
| | | Test Conditions |
| | Assessment | Equivalency of Input parameters |
| | | Output Comparison |
| **Applicability** | | Relevance of the quantities of interest |
| | | Relevance of the validation activities to the context of use |

modelling in the medical device industry. The VV-40 document describes a pathway that encompasses the following steps: (i) definition of the question of interest (QOI), which will be addressed by the computational model; (ii) the definition of the context of use (COU), which states the scope and role of the computational model in addressing the QOI; (iii) assessment of the model risk within the COU and the potential effects of adverse outcomes as a consequence of an incorrect decision driven by the model; (iv) establishment of credibility goal and rigour required for the verification, validation, and applicability activities such that the model credibility is commensurate with the model risk [39], as summarized in Table 1.

### 3.2 Study design

In this study, the credibility of FE modelling of a self-expanding Ni-Ti stent was addressed through conventional scenarios where structural analysis is employed for such devices (examples in **Table A1** in the S1 Appendix). Common applications of FE models of Ni-Ti stents could be classified as (i) modelling of standard mechanical bench tests (e.g. axial tension/compression and radial compression test [21, 27]), (ii) predictions of deployment either in an idealised straight vessel or in patient-specific realistic anatomies [15, 16], and (iii) predictions of fatigue behaviour [14]. These applications were classified in three COUs and relevant VV activities were selected accordingly:

- *COU-1*: *Radial compression* is often used for deriving the material parameters for Ni-Ti [40] or evaluating the functional performance of different stent designs through relevant mechanical parameters (e.g., radial strength, radial resistive force, chronic outward force, stress field induced in the device) [27], as advised in the ASTM F3067 standard [41]. For COU-1, calculation verification was carried out by examining mesh discretization, temporal discretization, and element integration selection, while validation of the model inputs was performed through a detailed sensitivity analysis on both geometric and material model parameters.

- *COU-2*: *Stent deployment* predictions in vessel-like geometries are often used to establish device performance in restoring vessel patency, or assess the level of stress induced in the device or/and in the vessel tissue [16, 42]. For COU-2, calculation verification was carried out for mesh discretization, temporal discretization, and element formulation selection, while validation of the computational model was investigated through *in vitro* comparators.

**Table 2. Definition of the COUs and the related verification and validation activities addressed within this study.**

|  | COU-1 | COU-2 | COU-3 |
|---|---|---|---|
| Context Of Use | Predict the mechanical performance of the stent subjected to a radial compression test. | Predict the configuration of a stent when deployed in a vessel, considering idealized and realistic geometries. | Predict the fatigue performance of the device when subjected to radial pulsatile pressure. |
| Quantity of Interest (QoI) | • Radial force—diameter curve and derived parameters such as maximum force at crimp ($F_{max}$) and chronic outward force (COF).<br>• Stress distribution on the peak of the strut. | • Vessel outer/inner diameter (OD/ID), diameter gain.<br>• Minimum lumen area (MLA)<br>• Incomplete strut apposition (ISA). | • Mean and alternating strain components.<br>• Fatigue safety factor (FSF). |
| Verification Activities | Aspects of calculation verification:<br>• mesh density;<br>• element integration;<br>• stable time increment. | Aspects of calculation verification:<br>• mesh density;<br>• element integration;<br>• stable time increment. | Aspects of calculation verification:<br>• mesh density;<br>• element integration;<br>• stable time increment. |
| Validation Activities | Sensitivity analysis on model inputs:<br>• geometry (strut thickness and width) considering a ±10% variation;<br>• nickel-titanium material parameters considering a ±10% variation. | Comparison with *in vitro* configuration of deployment:<br>• straight vessels made of silicone;<br>• 3D-printed patient-specific vessels made of resin. | Not addressed. |

- *COU-3*: *Fatigue predictions* from pulsatile loading of the device are commonly carried out to mimic ASTM F2477 [43] conditions. Simulations generally compute the mean ($\varepsilon_m$) and alternating strain ($\varepsilon_a$) components and evaluate the fatigue safety factor (FSF) to estimate lifetime of the device [44]. For COU-3, calculation verification was carried out for mesh discretization, temporal discretization, and element formulation selection to assess their effects on the prediction of $\varepsilon_m$ and $\varepsilon_a$ for the plotting of the constant life diagrams and on the FSF.

Table 2 provides a summary of the COUs under study, with details of the verification and validation activities performed.

### 3.3 General description of the FE model

**3.3.1 Stent device and functional unit.** The Zilver PTX (Cook Medical, USA) is a self-expanding Ni-Ti stent intended for the treatment of restenotic symptomatic lesions in the iliac artery, superficial femoral artery and in the above-the-knee femoropopliteal artery. The device is composed of sequential rings that feature open-cell Z-shaped struts with a peak-to-valley design and it is produced in a variable range of nominal diameter (6.0–10.0 mm) and length (20.0–200.0 mm) [45].

A 3D model of the device was created with Inventor® 2018 software (Autodesk, San Rafael, CA, United States) based on measurements from optical microscopy and micro-CT imaging. Similar to most stent designs, the Zilver PTX has a repeating pattern (Fig 1a) that allows a functional unit of the 6.0 × 60 mm device to be used to reduce the system size, with symmetric boundary conditions in the circumferential direction (θ) applied at the truncated edges (Fig 1c). This is a common approach for stent modelling where geometry, loading and boundary conditions possess symmetry [46, 47]. The radial response of the functional unit was quantitatively compared to the full stent to ensure this assumption was valid (details in the Supporting information file). The functional unit model was used for running verification activities to investigate the effects of discretization error, and for running the sensitivity analysis over model inputs. Otherwise, the full geometry of the stent was considered when validating against the *in vitro* mock vessel comparators, as it is more representative of the stent-to-lumen interaction along the whole length of the device. A stent with a length of 60.0 mm was modelled in

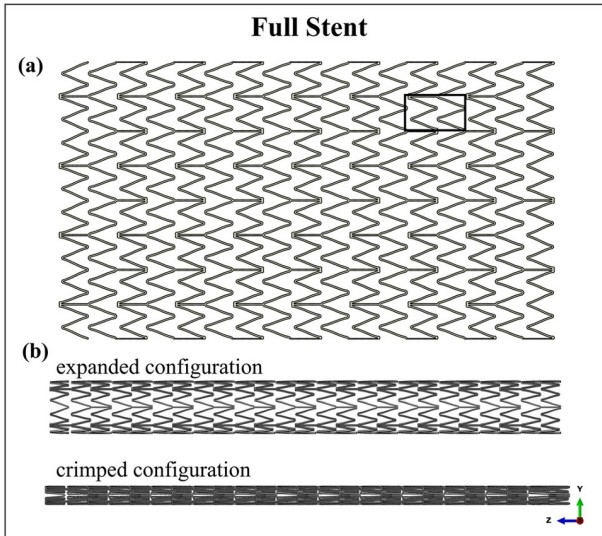
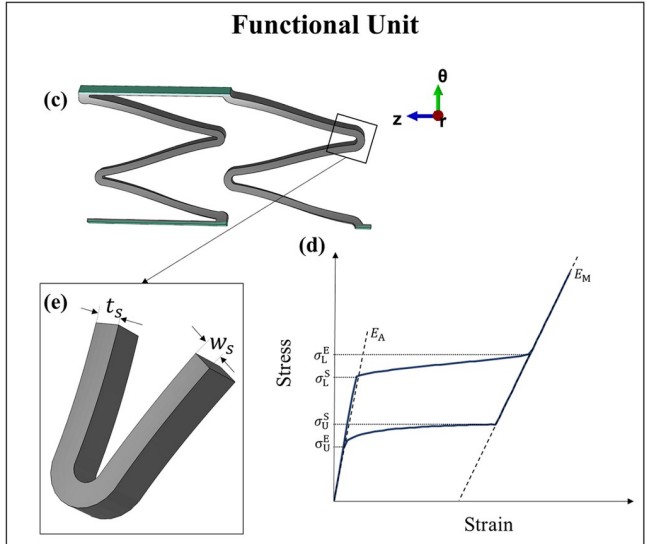

**Fig 1.** Stent model: **(a)** 2D sketch of the stent with detail of the functional unit used in the verification step; **(b)** comparison of crimped and expanded configurations; **(c)** functional unit; **(d)** superelastic Ni-Ti constitutive law; **(e)** details of width **($w_s$)** and thickness **($t_s$)** strut dimensions.

varying diameter sizes (6.0, 7.0 and 9.0 mm) to match the devices available for the experimental campaign.

The complex nonlinear behaviour of Ni-Ti (Fig 1d) was implemented through the built-in VUMAT material formulation by Auricchio and Taylor [31] available in Abaqus, where the values for material parameters were determined from experimental tests in a previous work [41].

**3.3.2 Simulation strategy and settings.** The FE model included non-linearity from the superelastic material constitutive law of Ni-Ti [31], the complex contact interactions and large deformations occurring in the analysis. The Abaqus/Explicit solver (v. 2017, SIMULIA, Dassault Systèmes) was used to solve the dynamic equations of equilibrium [46], whose general form for a non-linear explicit analysis is written as:

$$[M]\{\ddot{u}\} + [C]\{\dot{u}\} + [K]\{u\} = \{F(t)\} \tag{3.1}$$

where $\{F(t)\}$ is the time-dependent load vector; $[M]$, $[C]$ and $[K]$ are the global mass, damping and stiffness matrices; $\{\ddot{u}\}$, $\{\dot{u}\}$, $\{u\}$ are the nodal acceleration, velocity and displacement vectors, respectively [48, 49].

The steps implemented for the simulation of each COU are outlined here.

COU-1: A radial pattern of rigid plates was used to enforce compression of the device from a diameter of 6.00 mm to 2.4 mm within a time step of 4 s (Fig 2, Step 1). The displacement was then released within a time step of 4 s and to allow the stent to recover freely (Fig 2, Step 2). Interactions between the stent surface and the plates was active, as well as self-contact among the stent surface. For COU-1, a global and a local output were compared, namely the radial force-diameter curve and the trend of maximum principal stress at the internal edge of the V peak (Fig 3b).

COU-2: For implantation within a vessel, these models followed the identical process outlined for COU-1, whereby the stent was crimped from a diameter of 6.0 mm to 2.4 mm (Fig 2, Step 1) followed by release in the same timeframe. However, during the release step in

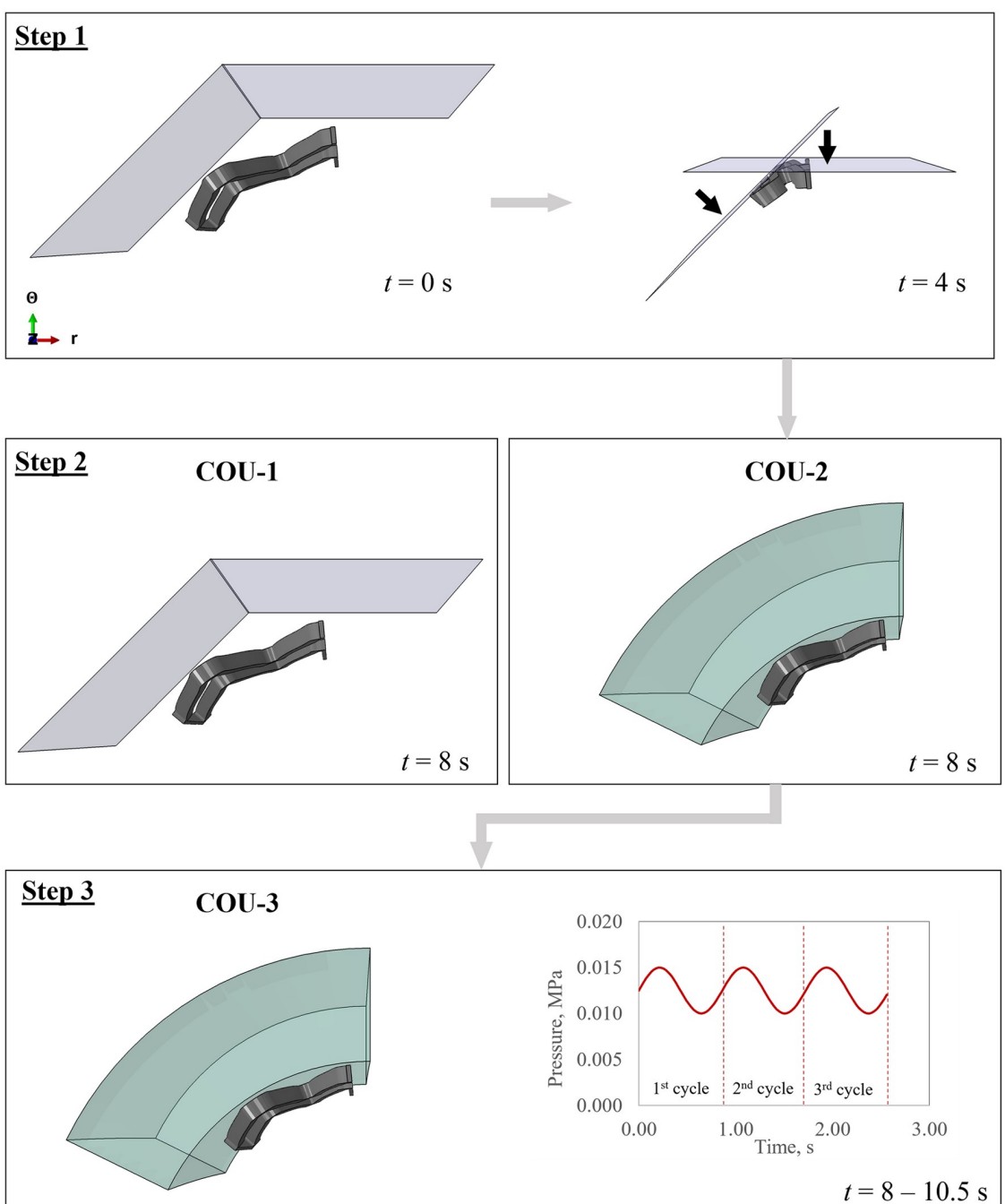

**Fig 2. Description of the steps for the simulations. Step 1** includes the crimping of the sub-unit, which is either allowed a free release (COU-1) or deployed in contact with the vessel (COU-2) in **Step 2**. Finally, the deployed configuration is used to apply pulsatile blood conditions in **Step 3** for fatigue analysis (COU-3).

COU-2, the interaction between the inner lumen of the vessel and the device was active to allow for deployment in both straight and patient-specific geometries (Fig 2, Step 2). For deployment in more realistic anatomical vessel an additional morphing procedure is required, as described in a previous paper [37]. For COU-2, clinically relevant parameters

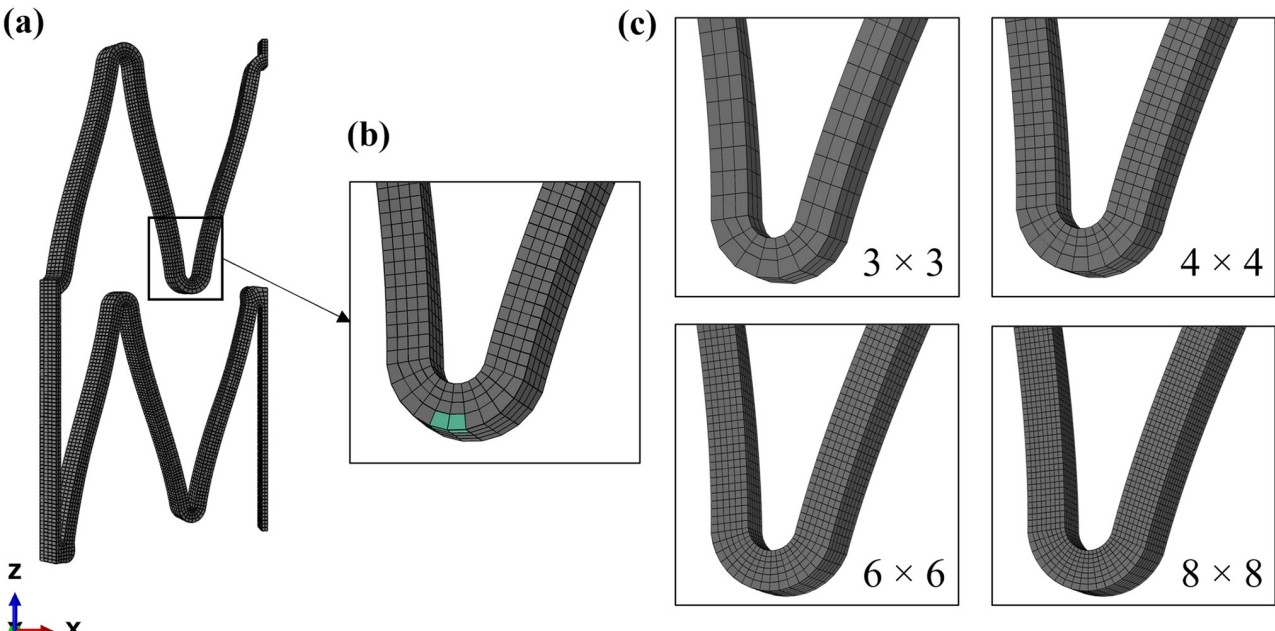

**Fig 3.** Mesh density: **(a)** uniform mesh of the functional stent unit; **(b)** details of the portion selected for the local QoI; **(c)** comparison of four different element densities.

such as the outer vessel diameter, minimum lumen area, and incomplete stent apposition were measured (**Fig 5c**).

COU-3: Following the implantation procedure described above, COU-3 included an additional time step to apply a pulsatile load to evaluate the fatigue behaviour. Three cycles of blood pressure oscillating between the diastolic and the systolic values of 80 mmHg and 120 mmHg with a frequency of 1.2 Hz, corresponding to the average blood pulsation conditions [23], were applied (Fig 2, Step 3). A strain-based criterion was chosen for the analysis of fatigue given the nature of Ni-Ti alloy constitutive law [14]. The alternating ($\varepsilon_a$) and mean strain ($\varepsilon_m$) components were computed with the scalar method [44], which was deemed suitable [8] for the type of loading conditions investigated:

$$\varepsilon_m = \frac{\varepsilon_{\text{systole}} + \varepsilon_{\text{diastole}}}{2} \tag{3.2}$$

$$\varepsilon_a = \frac{|\varepsilon_{\text{systole}} - \varepsilon_{\text{diastole}}|}{2} \tag{3.3}$$

The values of these components allowed to plot the constant-life diagram and the alternating strain, and were used to compute the fatigue safety factor (*FSF*) against the limit value for $10^7$ cycles $\varepsilon_a^* = 0.4\%$ [50], as follows:

$$FSF = \frac{\varepsilon_a^*}{\varepsilon_a} \tag{3.4}$$

During all the steps, the general contact algorithm with a hard contact definition and a friction coefficient of $\mu = 0.2$ was used, consistent with studies found in literature [13, 15, 51].

**Table 3. Characteristics of mesh grids.**

| Mesh Grid | 3 × 3 | 4 × 4 | 6 × 6 | 8 × 8 |
|---|---|---|---|---|
| Number of elements | 2,391 | 8,404 | 25,818 | 62,136 |
| Number of nodes | 4,348 | 13,495 | 35,910 | 79,929 |
| Element characteristic length, mm | 6.6E-02 | 4.90E-02 | 3.3E-02 | 2.5E-02 |
| Average aspect ratio, - | 2.14 | 1.85 | 1.80 | 1.82 |

## 3.4 Verification activities

Most of the studies available in the literature rely on commercial software whose code verification is extensively documented by their developer, however it would be advisable to run benchmark simulations to ensure the software produces accurate results on local hardware platforms [52, 53]. Therefore, the verification activities we have focused on here are related to the calculation verification, where we specifically address aspects related to the spatial and temporal refinement [18] of the system, as follows:

- Mesh density (*MD*): to quantify the error associated with spatial discretization and mesh quality.

- Element integration (*EI*): to address the influence of different mathematical formulations in the derivation needed to deduce the displacements of finite elements [54].

- Target time increment (*TTI*): to determine the acceptable stable time increment in the dynamic simulation.

**3.4.1 Mesh density.** Four different mesh refinements with a density of elements in the width and thickness of the strut of 3 × 3, 4 × 4, 6 × 6, and 8 × 8 elements (Fig 3) were selected. Details regarding the number of elements and nodes, characteristic length and aspect ratio are reported in Table 3.

**3.4.2 Element integration.** Three-dimensional linear solid elements were used, whose formulation is available with full or reduced integration options. Within the reduced formulation, the kinematic split is selected as *average* (default), while *orthogonal* and *centroid* are proposed as alternatives to decrease the computational costs [55]. The enhanced hourglass control option was added and compared. The 4 × 4 mesh grid was used and details of all the simulation runs are reported in Table 4.

**Table 4. Characteristics of element integration.**

| Element Integration | Code | Label | Description |
|---|---|---|---|
| Full integration | C3D8 | Full | 8 points to integrate the polynomial terms in an element's stiffness matrix. |
| Reduced integration | C3D8R | | 1 point to integrate the polynomial terms in an element's stiffness matrix. Different options arise with the choice of kinematic split: |
| | | Red-A | **average:** based on the uniform strain operator |
| | | Red-AH | **average + enhanced hourglass control** |
| | | Red-O | **orthogonal:** based on the centroidal strain operator and a slight modification to the hourglass shape vector |
| | | Red-C | **centroid:** the centroidal strain operator and the hourglass base vectors |

**3.4.3 Time increment.** The Abaqus/Explicit solver was chosen for the analysis due to the complex nonlinear contact conditions in the deployment simulation. The explicit method is conditionally stable since there exists a critical time step $\Delta t_{min}$ that must not be exceeded in the analysis and depends on the highest eigenvalue ($\omega_{max}$) in the model and the fraction of critical damping ($\xi$) in the highest mode [56]:

$$\Delta t_{min} \leq \frac{2}{\omega_{max}} \left( \sqrt{1 + \xi^2} - \xi \right) \tag{3.5}$$

The explicit procedure solves the system as a wave propagation problem, moving at a dilatation speed of $c_d$, which for a linear elastic material with a Poisson's ratio of zero depends on Young's modulus $E$ and the material density $\rho$:

$$c_d = \sqrt{\frac{E}{\rho}} \tag{3.6}$$

The magnitude of the critical time step depends on the largest natural frequency of the linearized system, hence on the size of the smallest characteristic element length $L^e$ and the dilatation wave speed $c_d$. The stability time limit can be estimated by:

$$\Delta t = \frac{L^e}{c_d} \tag{3.7}$$

To efficiently model a quasi-static problem where the material behaviour is rate-independent (e.g. no viscous behaviour), the loading rate can be increased and mass scaling applied [16, 21–23, 27, 57]. Typically, simulations are understood to be quasi-static by evaluating the ratio between kinetic to internal energy ratio [21, 23] and ensuring that this ratio is below 5% for the simulation. While this criterion appears widely in the literature, it has rarely been independently assessed. In this study, a fixed mass scaling was applied at the beginning of the step on the stent specified by selecting an element target time increment (*TTI*) with values of 1.0E-05 s, 5.0E-06 s, 1.0E-06 s, 5.0E-07 s for a total step time of 8 s, on the 4 × 4 mesh grid. Results obtained through the explicit solver for the range of *TTI* were compared with a simulation run with the dynamic implicit method. The implicit algorithm requires iterative solutions for each time increment, thereby it ensures accuracy of the solution dictated by the convergence criterion [58]. A summary of the analyses for calculation verification activities is reported in Table 5.

## 3.5 Validation activities

The validation activities assessed the degree to which the computational model was able to physically represent the quantities of interest (QoI) in the intended uses [18]. In this study, we sought to understand the sensitivity of computational model inputs on the predicted result, while also comparing predictions directly to an *in vitro* comparator. For all the validation activities presented here, the optimum model parameters were chosen based on the detailed verification analysis, which meant that all validation models had a mesh density of 4 × 4, a reduced integration element formulation (Red-A) and a target time increment of 5.0 E-6 s.

**3.5.1 COU-1 validation—Model input analysis.** The geometry and the material properties of the stent were considered key parameters for the sensitivity analysis on model inputs. For the sensitivity to geometrical inputs, strut thickness ($t_s$ = 0.197 mm) and width ($w_s$ = 0.120 mm) were considered. These dimensional measurements are often acquired with a variety of imaging techniques [14, 15, 19, 27, 59] and are subject to uncertainties. Even when the

**Table 5. Summary of the parameters for the calculation verification study.**

| Model | Mesh Density [width × thickness] | Element Integration | Time Increment [s] |
|---|---|---|---|
| *MD –3 × 3* | 3 × 3 | Full integration | 5.0E-06 |
| *MD– 4 × 4* | 4 × 4 | | |
| *MD– 6 × 6* | 6 × 6 | | |
| *MD– 8 × 8* | 8 × 8 | | |
| *EI–Full* | 4 × 4 | Full integration | 5.0E-06 |
| *EI–Red-A* | | Reduced—Average | |
| *EI–Red-O* | | Reduced—Orthogonal | |
| *EI–Red-C* | | Reduced—Centroid | |
| *TTI– 1.0E-05* | 4 × 4 | Full integration | 1.0E-05 |
| *TTI– 5.0E-06* | | | 5.0E-06 |
| *TTI– 1.0E-06* | | | 1.0E-06 |
| *TTI– 5.0E-07* | | | 5.0E-07 |

geometry is known (e.g., CAD is provided by the stent manufacturer), it typically represents the laser-cut configuration, which could be altered during subsequent electropolishing steps [60]. Similarly, to define the material parameters for Ni-Ti, experiments on material specimens have been carried out, although these might present variability among samples, lots, and suppliers [61]. In other cases, studies have used values from the literature, despite the fact that there is substantial variability in these parameters, depending on the alloy composition and/or processing history [61, 62]. In this study, the geometric thickness and width of the stent (Fig 1e) and six material parameters describing the Ni-Ti tensile behaviour (Table 6) were considered for the sensitivity analysis. As there is no information on the distribution of the error on these measurements, a ± 10% range is assumed, as advised in literature [63].

**3.5.2 COU-2 validation—*In vitro* comparator.** The validation for the COU-2 was performed to assess the deployment configuration of the self-expanding stent. We considered implantation both in a straight vessel to assess the capability of the model in predicting the stent-to-vessel interaction and final outer diameter (*OD*), and in 3D-printed patient-specific geometries to access quantitative data as the minimum lumen area (*MLA*) and the incomplete stent apposition (*ISA*), as reported in **Fig 5c.** The straight vessels were manufactured from silicone by Dynatek Labs, Inc. (Galena, MO, USA) to ensure consistency with the guidelines of ISO 7198 [64], while two patient-specific vessel geometries acquired in previous studies [37, 65, 66] were manufactured using a Form 3 Low Force Stereolithography 3D-printer (Formlabs, Massachusetts, USA). Informed consent statement was obtained for the patient-specific data, as reported in previous studies where these were used [37, 65, 66]. The studies were approved by the Institutional Review Board at the University of Florida and conformed to the

**Table 6. Nickel-titanium material parameters used as reference in the sensitivity analysis.**

| Parameter | Symbol | Reference Value |
|---|---|---|
| Austenite elasticity, MPa | $E_A$ | 47,000 |
| Martensite elasticity, MPa | $E_M$ | 20,000 |
| Start of transformation loading stress, MPa | $\sigma_L^S$ | 420 |
| End of transformation loading stress, MPa | $\sigma_L^E$ | 500 |
| Start of transformation unloading stress, MPa | $\sigma_U^S$ | 150 |
| End of transformation unloading stress, MPa | $\sigma_U^E$ | 115 |

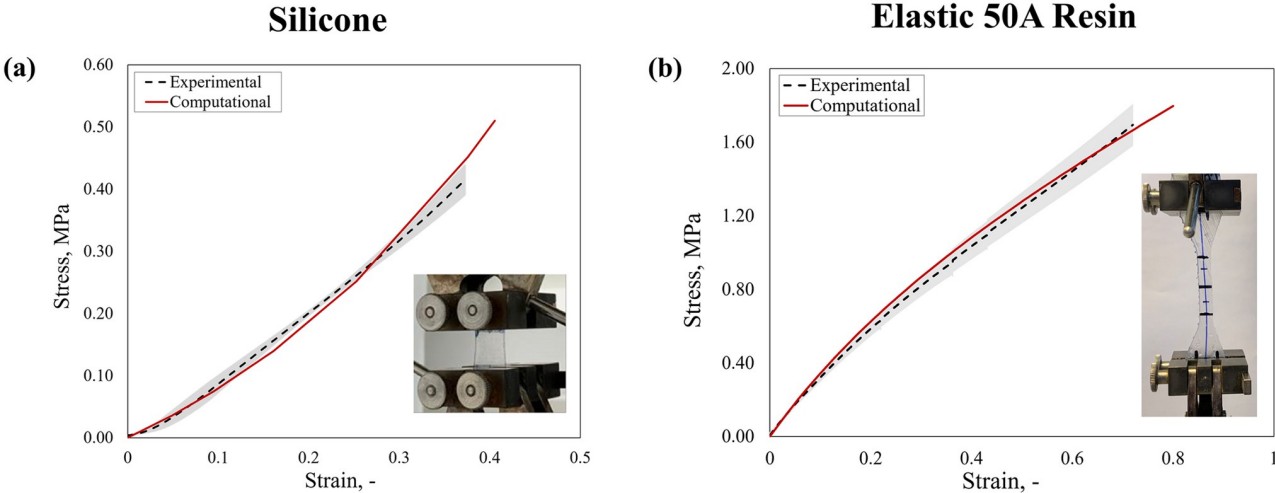

**Fig 4.** Calibration of materials used for the mock vessels: **(a)** experimental and computational stress-strain curves derived from the hoop tests on silicone material; **(b)** experimental and computational stress-strain curves derived from uniaxial tensile dog-bone specimens of Elastic 50A Resin.

Helsinki Declaration on human research of 1975, as revised in 2000. Written informed consent was obtained from the patients.

*3.5.2.1 Characterization and calibration of mock vessel material.* For robust validation, vessel material parameters were derived directly from the specimens, whereby the mechanical response of the silicon mock vessel was characterized through hoop tests ($n = 3$), as detailed in a previous study [37]. In the simulation, silicone was modelled with an isotropic elastic constitutive law with a Poisson's ratio $v = 0.3$ and an elastic modulus $E = 0.65$ MPa, with these parameters appropriately capturing the uniaxial tensile test data (Fig 4a). Similarly, the Elastic 50A Resin material (Formlabs, Massachusetts, USA), designed for medical models undergoing large deformations [67], was used for the 3D-printed patient-specific geometries and was characterized by uniaxial tensile testing of dog-bone specimens ($n = 10$) according to the ASTM D638-14 standard [68]. A first-order isotropic hyperelastic Neo-Hookean law with $C_{10} = 0.458$ MPa and $D_1 = 0$, provided the optimal fitting of the experimental curve (Fig 4b).

*3.5.2.2 Experimental deployment in mock vessels.* The experimental deployment was performed in a water bath at a temperature of 37˚C to ensure devices were in the appropriate superelastic range. In the straight silicone vessels, Zilver stents ($n = 2$) featuring a 7.0 mm diameter were deployed, and the final configuration was assessed qualitatively and quantitatively in terms of the final outer diameter (*OD*) of the vessel and compared against simulation results. In each patient-specific scenario, Zilver stents with a diameter of 6.0, 7.0 and 9.0 mm were implanted. This allowed two clinical scenarios to be evaluated across three oversizing conditions, which represented the extreme cases of under-sizing and oversizing, as well as a condition of normal oversizing within the range [37]. The configurations of the deployed stents were acquired using 3D micro-CT scanning (μCT-100, Scanco Medical, Switzerland) with an X-ray source of 70 kV and the Al filter of 0.5 mm. DICOM images from the micro-CT scans were compared against the cross-sectional images from the computational model through ImageJ software (National Institute of Health, USA) to quantify relevant clinical performance indicators, such as outer diameter (*OD*), minimum lumen area (*MLA*) and incomplete stent apposition (*ISA*) as illustrated Fig 5c.

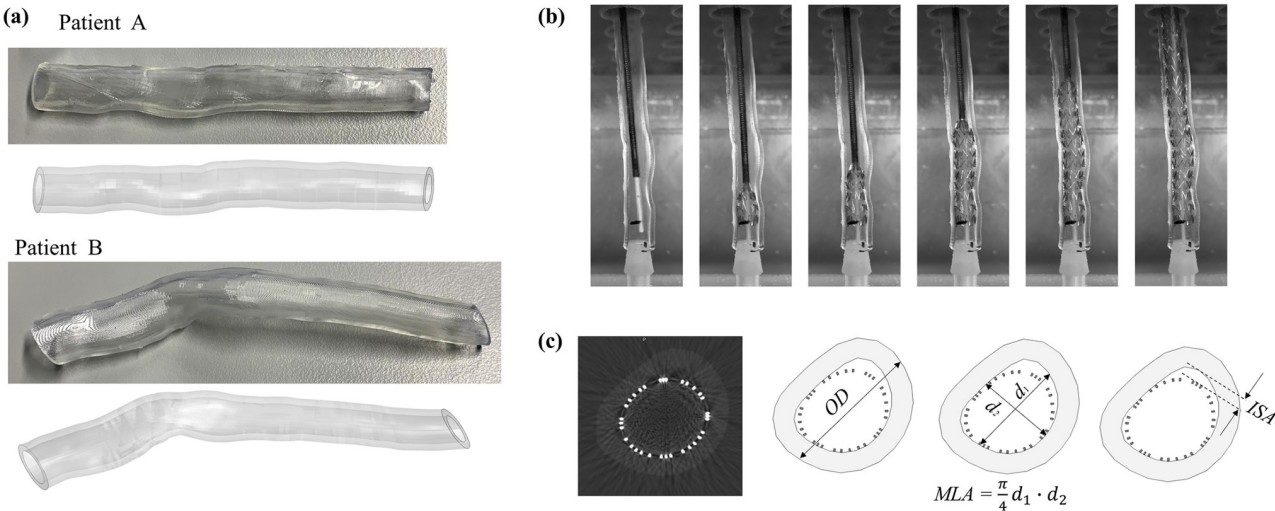

**Fig 5.** (a) Comparison of patient-specific geometries as 3D-printed resin vessels (top) and FE model (bottom); (b) sequence of stent deployment into a resin mock vessel; (c) DICOM cross-sectional image (left) and schematics of the performance indicators.

## 4 Results

### 4.1 COU-1

**4.1.1 COU-1 calculation verification in radial compression simulation.** *4.1.1.1 Mesh Density (MD)*. Global radial force and local maximum principal stresses of successive runs of the model with increasing mesh density were quantified (Fig 6) and compared to the prediction obtained with the $8 \times 8$ grid (Table 7). For the global output, the coarsest mesh ($3 \times 3$) showed the largest inaccuracy in predicting the value of chronic outward force (*COF*) with a deviation of -13.8%, while the other grid refinements predicted forces within ±2%. As for the local output, mesh grids either over- or under-predicted the maximum principal stress with a deviation below 2% when compared to the finest grid. Overall, with mesh refinement, the element characteristic length decreased and the number of elements in the grid increased, which contributed to larger computational times. An optimal compromise was found for a $4 \times 4$ grid, which gave a reduction of 82% of CPU time and global and local predictions with inaccuracy below 2%, comparable to the finest $8 \times 8$ mesh.

*4.1.1.2 Element Integration (EI)*. Global and local effects of different element integration options were assessed, with the results shown in Fig 7 and Table 8. The default option for reduced integration (Red-A) was considered as the reference model, as reduced integration is suggested in the Abaqus manual for hexahedral elements applied to bending problems (e.g., stent crimping and deployment) [17, 55]. For the global output, the different *EI* options resulted in an over-prediction of the force at maximum crimp in the range of +13.7% to +- 49.6% when compared to Red-A. Interestingly, lower discrepancies were found for the local output, where the over-prediction error was below +10%, except for the centroid option (Red-C) where a drop in maximum principal stress of -13.4% was reported. As for the computational time, reduced integration (Red-A, Red-O, Red-C) decreased the CPU time by 25% compared to full integration, however, the enhancement of hourglass control (Red-AH) increased the run time by 3-fold compared to default (Red-A).

*4.1.1.3 Target Time Increment (TTI)*. Global and local effects of different target time increments (*TTI*) were compared to the dynamic implicit solver, which uses the implicit time integration to calculate the transient dynamic or quasi-static response of a system, with results

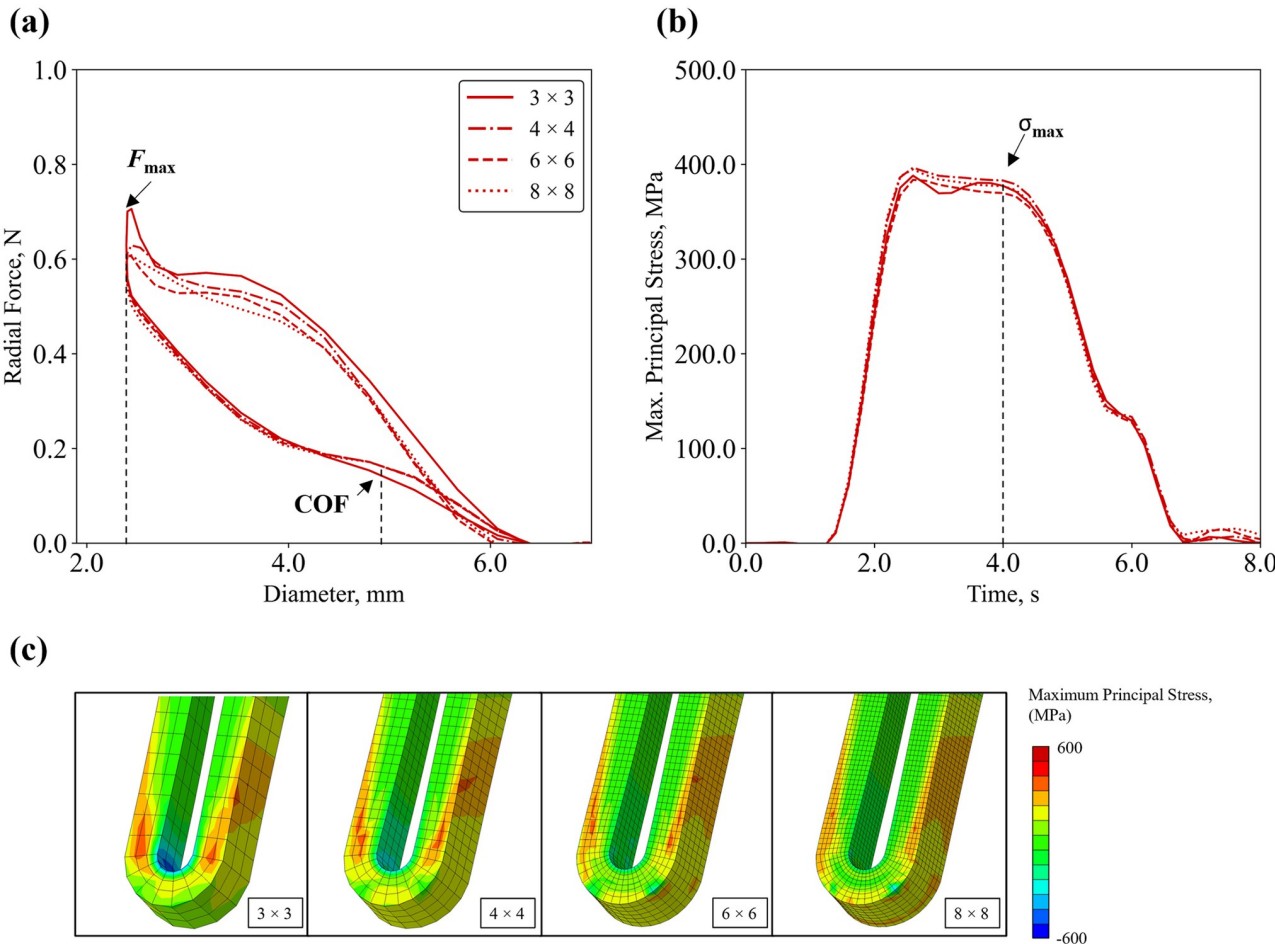

**Fig 6.** Calculation verification on mesh density for radial compression simulation: **(a)** radial force–diameter curves; **(b)** maximum principal stress–time curves; **(c)** contour plots of maximum principal stress for different mesh grids.

shown in Fig 8 and Table 9. For the global output, the *TTI* value had a minor influence on the radial force at maximum crimp, with a maximum variation of -9.5%. Similarly, the local output only showed small discrepancies, with the largest deviation of -4.0% compared to the dynamic implicit case. Overall, a *TTI* = 5.0E-06 s provided reliable results with optimum performance in terms of computing time. For all the analyses, a peak of the KE to IE ratio was found at the

**Table 7. Results of the calculation verification on mesh density applied to radial compression simulation.**

|  | Verification | | | |
|---|---|---|---|---|
| **Mesh Density** | **3 × 3** | **4 × 4** | **6 × 6** | **8 × 8** |
| CPU Time | 0.09C | 0.20C | 0.46C | C |
| Force at Maximum Crimp, N | 0.63 | 0.59 | 0.59 | 0.58 |
| Difference, % | 8.94% | 1.76% | 1.54% | / |
| Chronic Outward Force, N | 0.13 | 0.16 | 0.16 | 0.16 |
| Difference, % | -13.83% | 0.26% | -0.13% | / |
| Stress at Maximum Crimp, MPa | 377.27 | 382.90 | 369.69 | 376.93 |
| Difference, % | 0.09% | 1.58% | -1.92% | / |

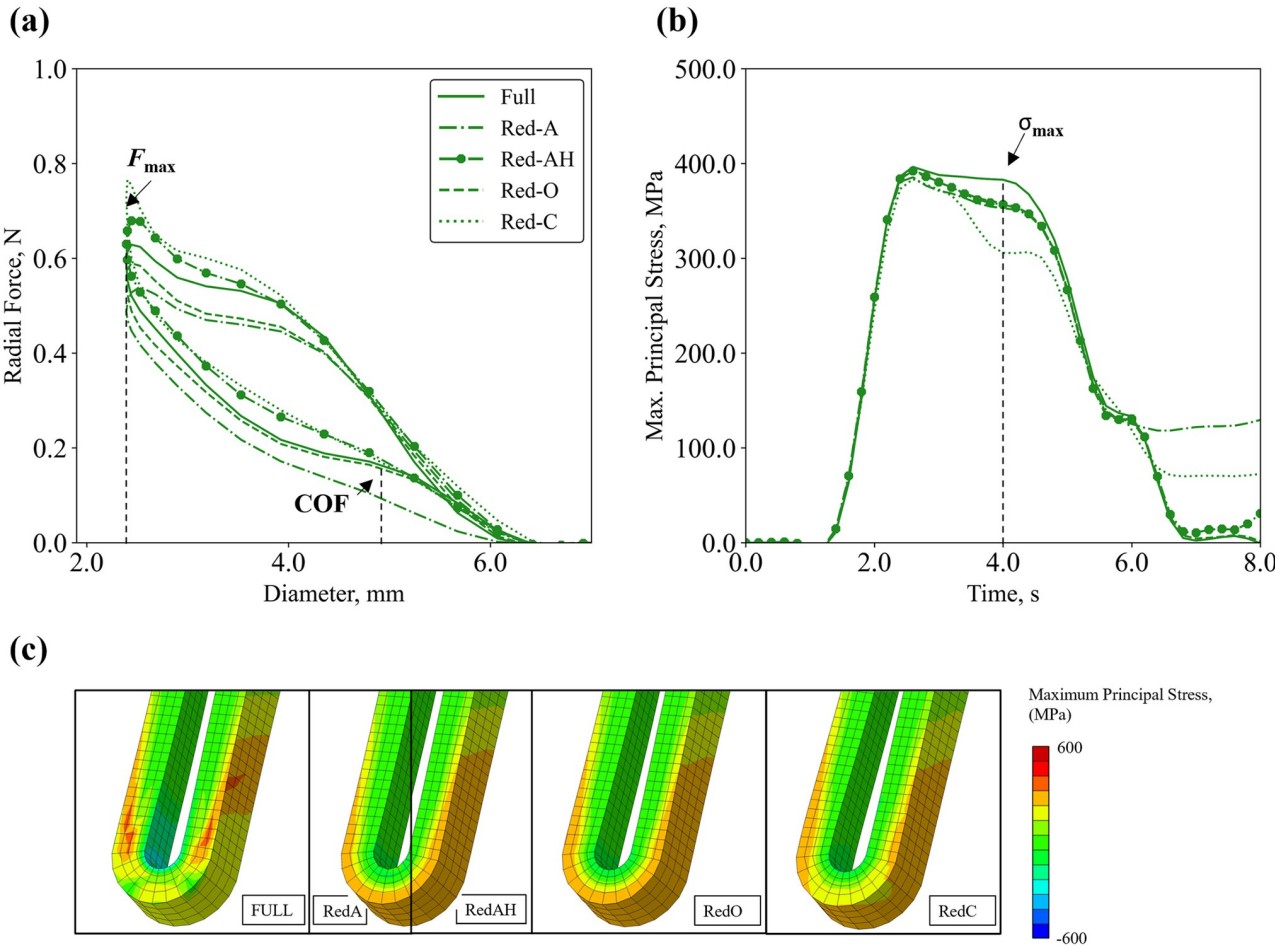

**Fig 7.** Calculation verification on element integration applied to crimping simulation using a mesh density 4 × 4: **(a)** radial force–diameter curves; **(b)** maximum principal stress–time curves; **(c)** contour plots of maximum principal stress.

beginning of the simulation when the plates and the stent first come into contact with one another, however, for the rest of the analyses this ratio was below 5% (Fig 8c), which is the suggested criteria to ensure quasi-static conditions.

**4.1.2 COU-1 validation: Sensitivity analysis on model inputs.** A sensitivity analysis of geometrical and material parameters of the Ni-Ti stent was carried out for the radial

**Table 8. Results of the calculation verification on element integration applied to radial compression simulation.**

| Element Integration | Verification | | | | |
|---|---|---|---|---|---|
| | **Full** | **Red-A** | **Red-AH** | **Red-O** | **Red-C** |
| CPU Time | 4.00C | C | 3.20C | C | C |
| Force at Maximum Crimp, N | 0.59 | 0.49 | 0.63 | 0.55 | 0.73 |
| Difference, % | 20.51% | / | 29.25% | 13.72% | 49.60% |
| Chronic Outward Force, N | 0.16 | 0.16 | 0.17 | 0.15 | 0.16 |
| Difference, % | 0.00% | / | 5.61% | -4.48% | 1.92% |
| Stress at Maximum Crimp, MPa | 382.90 | 352.95 | 356.70 | 358.11 | 305.77 |
| Difference, % | 8.48% | / | 1.06% | 1.46% | -13.37% |

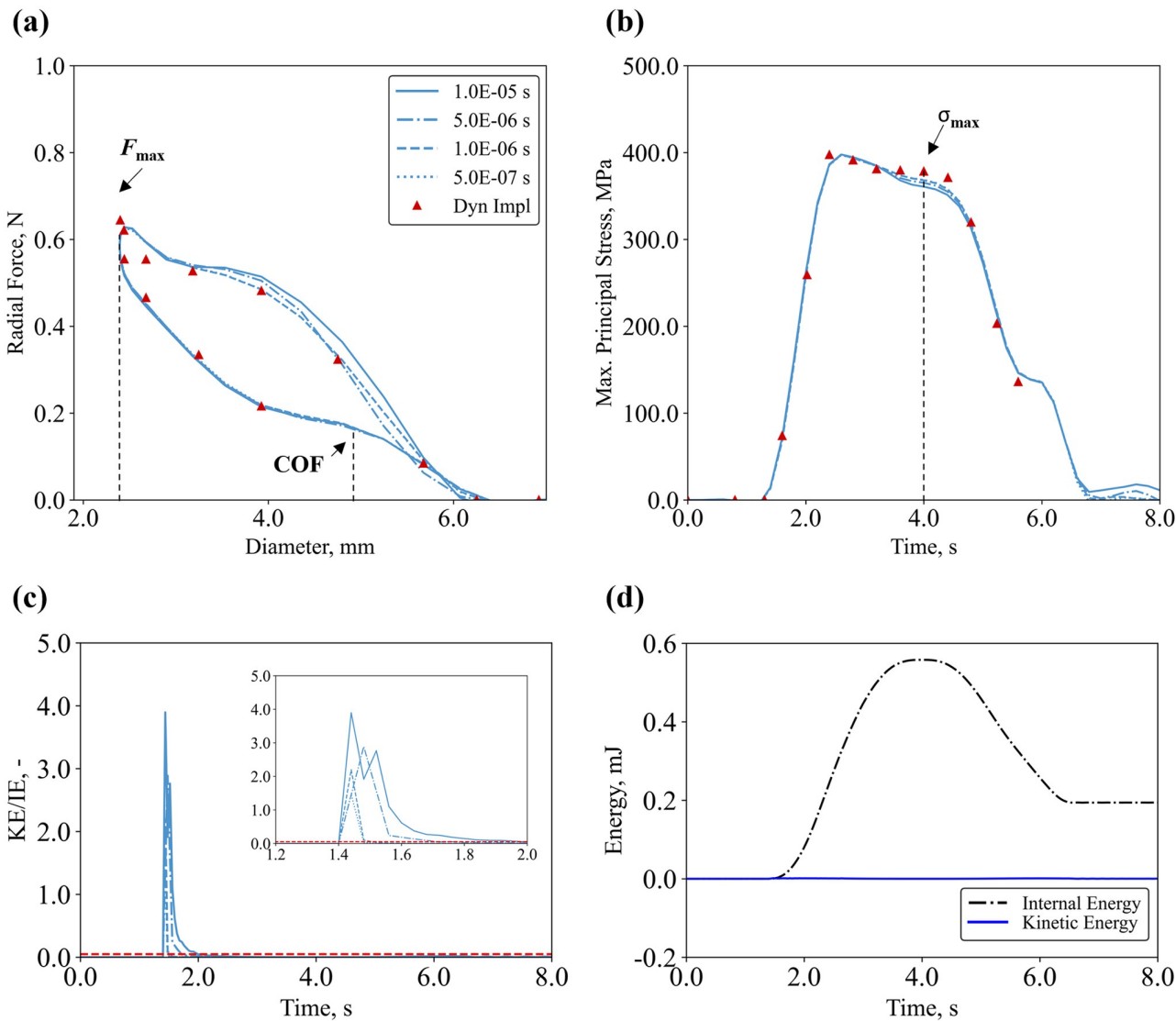

**Fig 8.** Calculation verification on target time increment applied to crimping simulation: **(a)** radial force–diameter curves; **(b)** maximum principal stress–time curves; **(c)** comparison of kinetic to internal energy ratio trends with the threshold limit of 5% (red dashed line); **(d)** comparison of kinetic and internal energies in the TTI = 5.0E-06 s case.

**Table 9. Results of the calculation verification activity on target time increment applied to crimping simulation.**

| | Verification | | | | |
|---|---|---|---|---|---|
| **Target Time Increment** | **Dyn Impl** | **1E-05 s** | **5E-06 s** | **1E-06 s** | **5E-07 s** |
| CPU Time | 20.53C | 0.89C | 0.89C | 0.91C | C |
| Force at Maximum Crimp, N | 0.64 | 0.58 | 0.59 | 0.59 | 0.59 |
| Difference, % | / | -9.47% | -8.87% | -8.51% | -8.66% |
| Chronic Outward Force, N | / | 0.16 | 0.16 | 0.16 | 0.16 |
| Difference, % | n/a | n/a | n/a | n/a | n/a |
| Stress at Maximum Crimp, MPa | 378.458 | 363.02 | 382.90 | 371.16 | 371.35 |
| Difference, % | / | -4.08% | 1.17% | -1.93% | -1.88% |

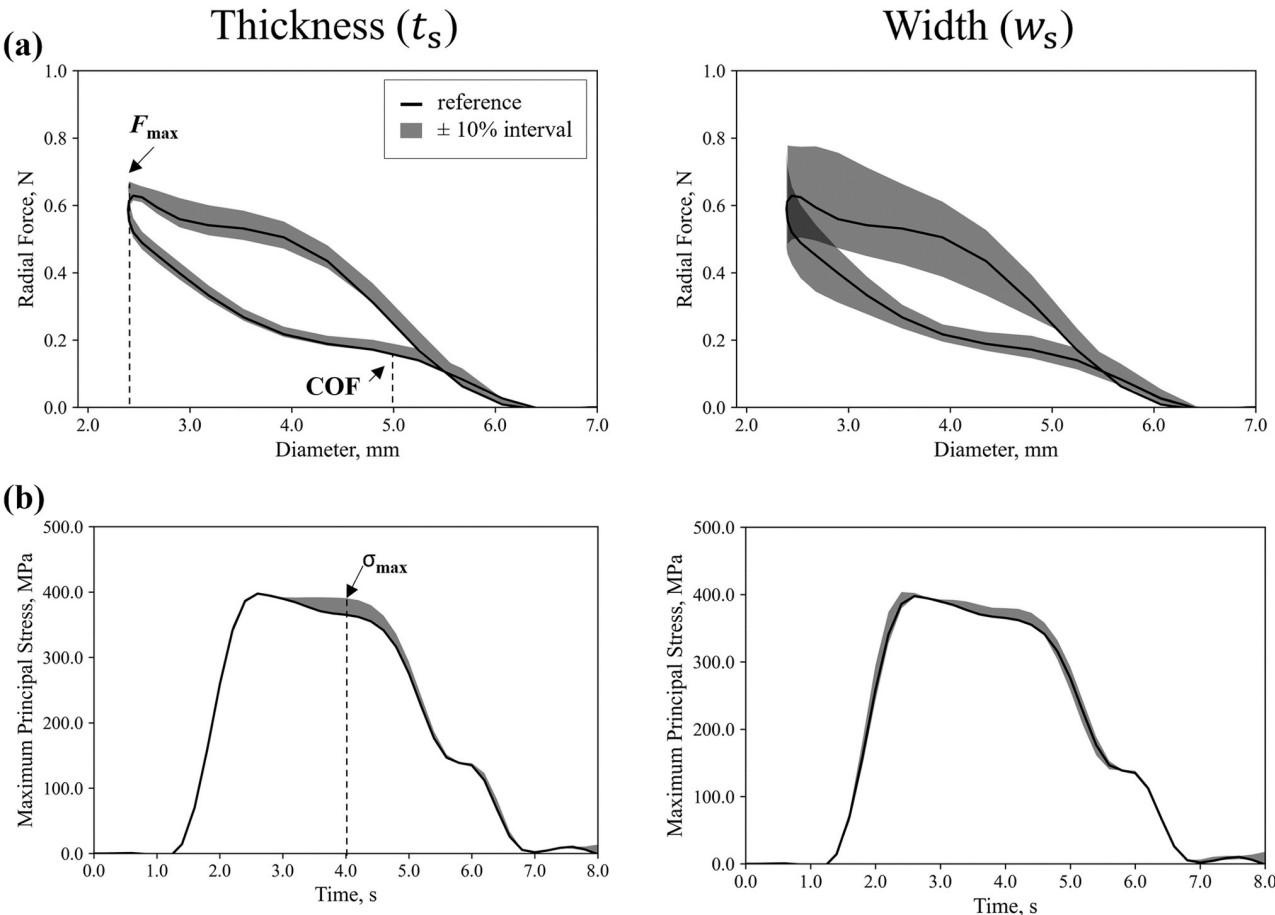

**Fig 9.** Model input sensitivity analysis on stent geometry: **(a)** effects on the global QoI; **(b)** effects on the local QoI when strut thickness and width are varied.

compression simulation (COU-1), whereby the effects on the global force and local stress parameters were quantified.

*4.1.2.1 Sensitivity analysis of stent geometry*. The sensitivity analysis of strut thickness and width on global and local outputs is reported in Fig 9 and Table 10. Greater effects were found for the radial force output, which consistently decreased or increased with respective decreases or increases in the strut dimensions. The variation of force at maximum crimp (-3.5%

**Table 10. Comparison of model input sensitivity analysis on stent geometry.**

| | Validation Model Inputs: Stent Geometry | | | | | |
|---|---|---|---|---|---|---|
| | Thickness ($t_s$) | | | Width ($w_s$) | | |
| | $t_s$—10% | $t_s$ | $t_s$ + 10% | $w_s$—10% | $w_s$ | $w_s$ + 10% |
| Force at Maximum Crimp, N | 0.59 | 0.61 | 0.66 | 0.47 | 0.61 | 0.76 |
| Difference, % | -3.52% | / | +8.90% | -22.18% | / | +24.82% |
| Chronic Outward Force, N | 0.16 | 0.16 | 0.18 | 0.13 | 0.16 | 0.20 |
| Difference, % | -0.89% | / | 16.10% | -16.64% | / | +24.99% |
| Stress at Maximum Crimp, MPa | 383.98 | 366.98 | 387.30 | 379.79 | 366.98 | 378.09 |
| Difference, % | +4.63% | / | +5.54% | +3.49% | / | +3.03% |

and +8.9%) and of chronic outward force (-0.9% and +16.1%) did not symmetrical decrease or increase with thickness. Reducing or increasing the width resulted in larger variations, both on the predicted force at maximum crimping (-22.2% and +24.9%) and of chronic outward force (-16.6% and +25.0%). Only minor effects on the maximum principal stress were found with the variation of the thickness ($< 5.5\%$) and width ($< 3.5\%$).

*4.1.2.2 Sensitivity analysis of nickel-titanium material parameters.* The sensitivity analysis of the Ni-Ti material parameters defined for the VUMAT superelastic behaviour [31] on global and local QoIs is reported are Fig 10 and Table 11. Comparing the force at maximum crimping, the highest variations were caused by the austenitic elastic modulus $E_A$ (-1.1% and +0.6%) and the start of transformation loading stress $\sigma_L^S$ (-4.5% to +4.9%), primarily affecting the loading curve. On the unloading portion of the curve, the chronic outward force at 5 mm diameter was mainly affected by $E_A$, $\sigma_L^S$ and $\sigma_U^S$ (maximum difference of +2.9%, +11.0%, -7.1% respectively), while $E_M$, $\sigma_L^E$ and $\sigma_U^E$ had marginal effect on the whole radial force curve.

Concerning the distribution of stress at maximum crimping, the effects of a variation of $E_A$ and $E_M$ were in the range ±1%. The highest effects were found for $\sigma_L^S$ which recorded a deviation in the range of ±9.5% while $\sigma_L^E$, $\sigma_U^S$ and $\sigma_U^E$ reported a variation less than ±1%.

## 4.2 COU-2

**4.2.1 COU-2: Calculation verification on deployment simulation.** For COU-2, the impact of mesh density (*MD*), element integration (*EI*) and target time increment (*TTI*) parameters on the predicted deployment configuration (e.g., inner lumen diameter, diameter gain) was analysed. The nodal coordinates of the vessel pre- and post-implantation were analysed with MATLAB® (v. 2021a, The Mathworks, Inc., Natick, MA) and the inner diameter was plotted against the normalized length of the vessel in Fig 11. Additionally, the average inner diameter (*ID*) and the corresponding diameter gain (*DG*), hereby defined as the difference between the inner diameter pre- and post- deployment $DG = ID_{post} - ID_{pre}$, were computed and compared with the reference simulation in Table 12.

Generally, the parameters accounted for calculation verification did not provide relevant variations for the prediction of the average inner diameter, with relative differences not larger than 2%. Greater differences were found when considering the diameter gain, with substantial discrepancies arising when 3 × 3 mesh (+17.5%) and *TTI* = 1.0E-05 s (+30.5%) were considered.

**4.2.2 COU-2: Validation with in vitro comparators.** *4.2.2.1 Deployment in a silicone straight mock vessel.* The validation of the stent-vessel interaction was achieved by qualitative and quantitative comparison of the final deployment in a straight silicone vessel. In the experiments, the outer diameter (*OD*) of the vessel was 7.2 mm prior to deployment and reached an average 7.57 mm and 7.62 mm after stent implantation. The comparison of *OD* prediction from the *in silico* model along the stent axis was shown in Fig 12a and reported a maximum error of 1.0% (Table 13).

*4.2.2.2 Deployment in 3D-printed patient-specific vessels.* The comparison between *in silico* and *in vitro* deployment in patient-specific vessels A and B is reported in Fig 13 for all stent sizes. The *in silico* model successfully predicted the final configurations both in a straighter geometry (A) and in curved/kinked geometry (B), capturing an improved apposition of the stent to the inner lumen for increasing diameter size and correctly identifying the locations along the vessel axis where strut-to-artery incomplete apposition occurred.

The quantitative comparison of the performance indicators is reported in Fig 14, where uncertainties arising from experiments were plotted as average and standard deviation over three measurements. The outer diameter (*OD*) was assessed in the proximal, central and distal

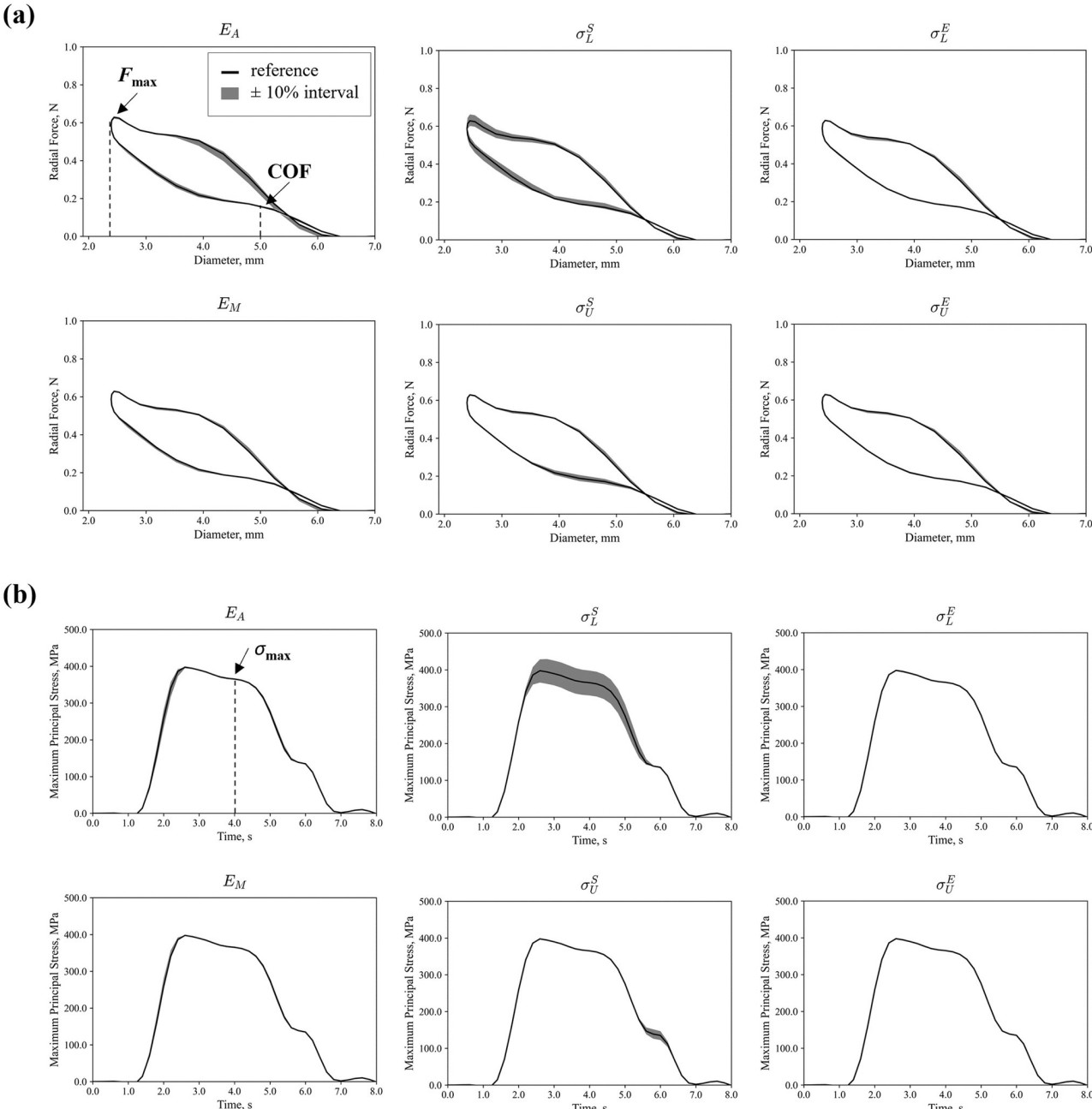

**Fig 10.** Sensitivity analysis on Ni-Ti material parameters: **(a)** effects of each parameter on the radial force; **(b)** effects of each parameter on the maximum principal stress at a V-peak.

portions of the vessel (*OD* was proposed as the metric for validation here as it could easily be measured by optical methods, should micro-CT not be available). The *in silico* model was successful in predicting the *OD* both for geometries A and B (Fig 14a and 14b) with an overall error smaller than +6.5%, although the level of accuracy depended on the local geometry and the vessel portion considered (Table 14). As for geometry A, the lowest accuracy was found in the distal region with the 9.0 mm device, while for B, the lowest accuracy was found in the central region with the 7.0 mm device (+5.0%). The minimum lumen area (*MLA*) was evaluated

**Table 11. Results for model inputs sensitivity analysis on nickel-titanium material parameters: Values and errors reported for the radial force curve and the maximum principal stress at the maximum crimp configuration.**

| | | Validation Model Inputs: Nickel-Titanium Parameters | | | | | |
| --- | --- | --- | --- | --- | --- | --- | --- |
| | | Force at Maximum Crimp, N | | | Stress at Maximum Crimp, MPa | | |
| | | Ref.—10% | Ref. | Ref.+ 10% | Ref.—10% | Ref. | Ref.+ 10% |
| $E_A$ | Value | 0.58 | 0.59 | 0.59 | 366.96 | 365.24 | 363.45 |
| | Difference, % | -1.05% | - | +0.61% | +0.47% | - | +0.00% |
| $E_M$ | Value | 0.58 | 0.59 | 0.59 | 364.99 | 365.24 | 363.45 |
| | Difference, % | -0.74% | - | +0.61% | -0.07% | - | 0.00% |
| $\sigma_L^S$ | Value | 0.56 | 0.59 | 0.62 | 330.53 | 365.24 | 398.00 |
| | Difference, % | -4.52% | - | +4.86% | -9.50% | - | +9.51% |
| $\sigma_L^E$ | Value | 0.59 | 0.59 | 0.59 | 363.53 | 365.24 | 365.52 |
| | Difference, % | -0.13% | - | +0.31% | -0.47% | - | +0.57% |
| $\sigma_U^S$ | Value | 0.59 | 0.59 | 0.59 | 364.96 | 365.24 | 364.96 |
| | Difference, % | -0.13% | - | -0.13% | -0.08% | - | +0.42% |
| $\sigma_U^E$ | Value | 0.59 | 0.59 | 0.59 | 364.96 | 365.24 | 364.96 |
| | Difference, % | -0.13% | - | -0.13% | -0.08% | - | +0.41% |

by considering the minimum inner diameter along the vessel centreline. The *in silico* model predicted the *MLA* in the geometries A and B (Fig 14c) with the highest error of +17.4% for the 9.0 mm device in geometry A and +3.7% for the 7.0 mm device in geometry B.

Incomplete stent apposition (*ISA*) was reported for the ring displaying the largest spacing from the arterial wall, with comparisons between *in silico* models and *in vitro* comparators shown in Fig 14. The model correctly predicted the location of the ring most detached from the wall, corresponding to the 9th and 2nd rings in A and B respectively (black arrows in Fig 13). However, deviations on the prediction of *ISA* magnitude were found, with the larger difference (-1.35 mm) reported for the 6.0 mm case in geometry B. Despite the deviation in the *ISA* value, the *in silico* model correctly predicted the presence (Y) or absence (N) of malapposition (whose threshold is defined as the detachment greater than strut thickness [69]) in all the cases, except for the 9.0 mm device in geometry B, where it overestimated the detachment distance.

## 4.3 COU-3

**4.3.1 COU-3: Calculation verification on fatigue behaviour simulation.** The effects of mesh density (*MD*), element integration (*EI*) and target time increment (*TTI*) on simulations of fatigue performance were assessed. The constant-life diagrams predicting the fatigue life of the devices are depicted in Fig 15, while the mean strain ($\varepsilon_m$) and alternating strain ($\varepsilon_a$) components, as well as the fatigue safety factor (*FSF*) are reported in Table 15. For the $\varepsilon_m$, value the 97.5th percentile was considered, as it gives a description of mean strain distribution and excludes peak values that might arise due to artificial local concentration [37]. Substantial differences were observed due to mesh density, with the 3 × 3 and 6 × 6 grids under- and over-predicting the mean strain distribution, with the 4 × 4 closer to the finer grid estimate. In the case of element integration, it was found that Full and Red-C tended to over-estimate the mechanical response compared to the default settings (Red-A). It was also found that as the *TTI* was decreased, the value of $\varepsilon_m$ increased.

For the $\varepsilon_a$, the maximum value was considered and used in the evaluation of the fatigue safety factor (*FSF*) as per Eq (3.4), which was then normalized with respect to the reference. While in all cases the model is far from predicting failure, differences were found. Among the

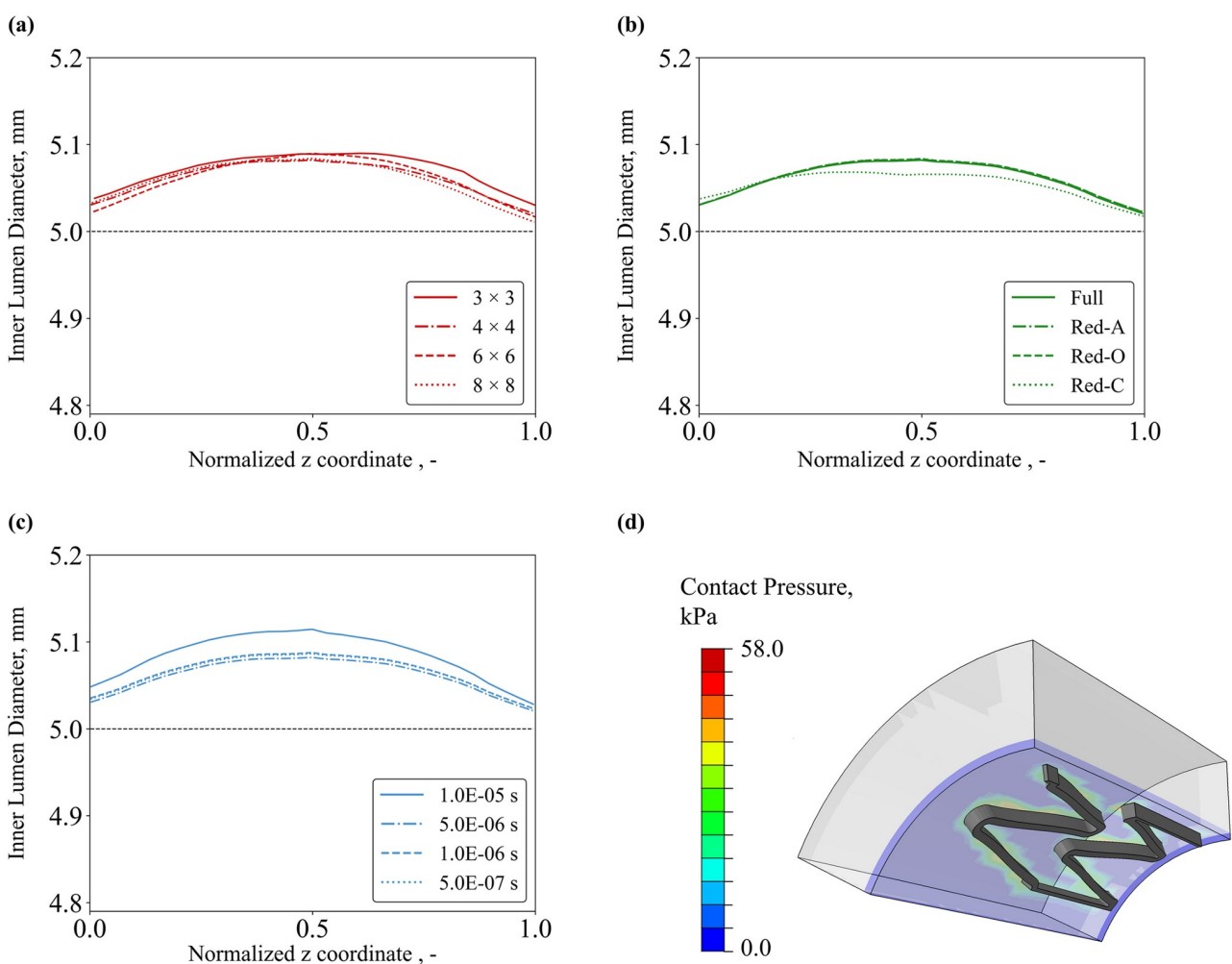

**Fig 11.** Calculation verification on deployment simulation: effects on prediction of inner lumen diameter due to **(a)** mesh density, **(b)** element integration and **(c)** target time increment; **(d)** contact pressure generated in the vessel due to stent expansion.

mesh grids, coarser refinement tended to over-predict the fatigue life behaviour, with the *FSF* obtained from the $4 \times 4$ grid being 6.5 times higher than the estimated *FSF* from the $8 \times 8$ grid. The choice of element integration did not alter the prediction of fatigue behaviour greatly, except for the Red-O option, where the prediction of *FSF* was approximately halved. The *TTI* substantial affected the *FSF*, with a 14.3-fold increase for smaller *TTIs*, as opposed to the largest stable increment time. This was mainly due to the substantial effect that *TTI* had on the prediction of the alternating strain component.

## 5. Discussion

### 5.1 General

While finite element modelling is now widely used to support the design and testing phases of nickel-titanium (Ni-Ti) cardiovascular devices [15, 16, 21, 22], there has been a lack of consensus on the methodology to assess model credibility for decision-making [39]. The ASME VV-40 documentation [11] provides a risk-informed credibility assessment of numerical models for medical devices by outlining a general framework for verification and validation activities

**Table 12. Average inner diameter and diameter gain for different mesh density, element integration and target time increment.** The column highlighted is taken as reference for evaluating the difference.

| | Verification on Deployment Simulation | | | |
|---|---|---|---|---|
| **Mesh Density** | **3 × 3** | **4 × 4** | **6 × 6** | **8 × 8** |
| Average Inner Diameter, mm | 5.07 | 5.06 | 5.06 | 5.06 |
| Difference, % | 0.21% | 0.02% | 0.04% | - |
| Diameter Gain, mm | 0.071 | 0.062 | 0.062 | 0.060 |
| Difference, % | +17.53% | +1.95% | +3.01% | - |
| **Element Integration** | **Full** | **Red-A** | **Red-O** | **Red-C** |
| Average Inner Diameter, mm | 5.06 | 5.06 | 5.06 | 5.05 |
| Difference, % | -0.02% | - | 0.00% | -0.17% |
| Diameter Gain, mm | 0.062 | 0.063 | 0.063 | 0.054 |
| Difference, % | -1.81% | - | +0.32% | -13.42% |
| **Target Time Increment** | **1E-05 s** | **5E-06 s** | **1E-06 s** | **5E-07 s** |
| Average Inner Diameter, mm | 5.09 | 5.06 | 5.07 | 5.07 |
| Difference, % | 0.39% | -0.07% | 0.02% | 0.00% |
| Diameter Gain, mm | 0.085 | 0.062 | 0.066 | 0.065 |
| Difference, % | +30.54% | -5.71% | +1.35% | - |

for numerical models. In this study, we addressed relevant verification and validation activities for finite element modelling of self-expanding Ni-Ti stents. Following a detailed literature review (**Table A1** in the S1 Appendix), three different contexts of use (COUs) were considered, namely (i) *Radial Compression* (ii) *Device Deployment* and (iii) *Fatigue Life Estimates*, and several quantities of interest (QoIs) were addressed for each. Generally, it was found that the chosen numerical parameters influenced global and local QoIs across all COUs considered, although to varying levels of sensitivity. It was found that both global and local quantities were highly sensitive to the mesh discretisation, with substantial variability in QoIs when the mesh

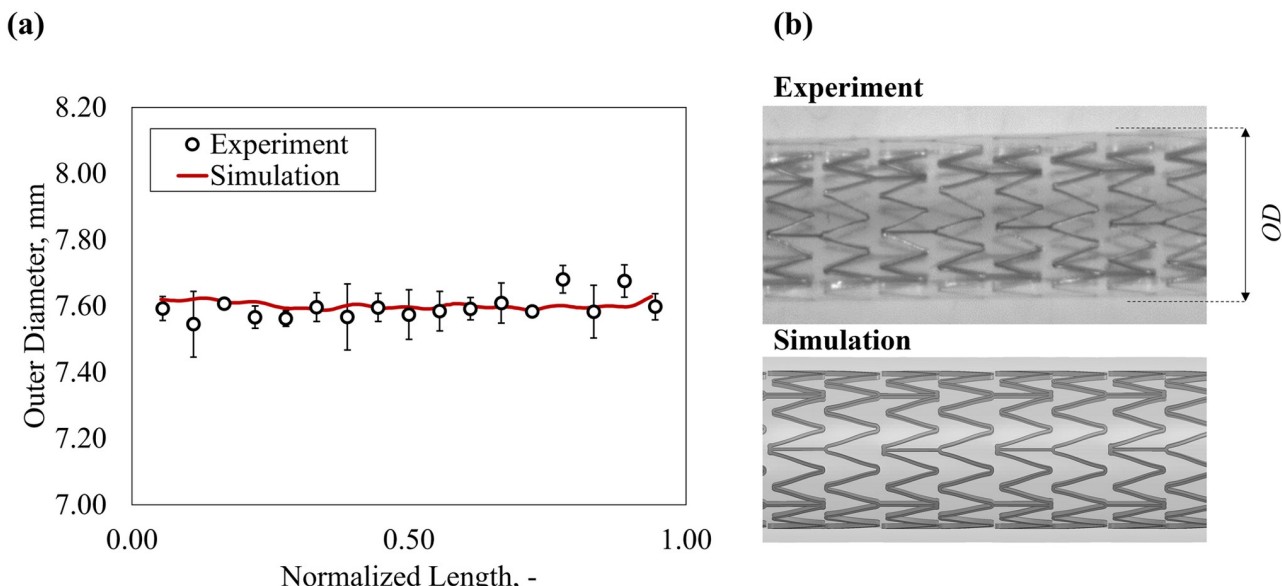

**(a)** **(b)**

**Fig 12.** Validation of the deployment in the straight silicone mock vessel: **(a)** measurements of the outer diameter (OD); **(b)** qualitative comparison of the deployed configuration.

**Table 13. Results of validation with straight mock vessel in vitro comparator.**

| Validation of Deployment: *in vitro* Straight Silicone Vessels | | | |
|---|---|---|---|
| Outer Diameter, mm [average ± SD] | | Model Error, % | |
| Experiment | Simulation | Maximum Error | Average Error |
| 7.59 ± 0.06 | 7.60 ± 0.07 | 1.04% | 0.40% |

density was less than 4 × 4 elements across the strut cross-section, while the element formulation also led to substantial variation depending on the chosen integration options. Furthermore, model results were generally highly sensitive to the chosen target time increment, irrespective of whether the ratios of kinetic and internal energies were kept below 5%. On the other hand, it was observed that model results showed relatively low sensitivity to Ni-Ti material parameters, which suggests that the calibration approaches used in the literature to date appear reasonable. While these general trends were observed, it was found that the sensitivity of the model predictions greatly depended on the COU being considered. Most notably, it was found that fatigue life estimates of Ni-Ti stents were extremely sensitive to the selection of model parameters. It was concerning that the estimated fatigue safety factor (*FSF*) could vary by almost an order of magnitude depending on the selection of model parameters, with the target time increment (*TTI*) being one of the most influential parameters. In contrast, the

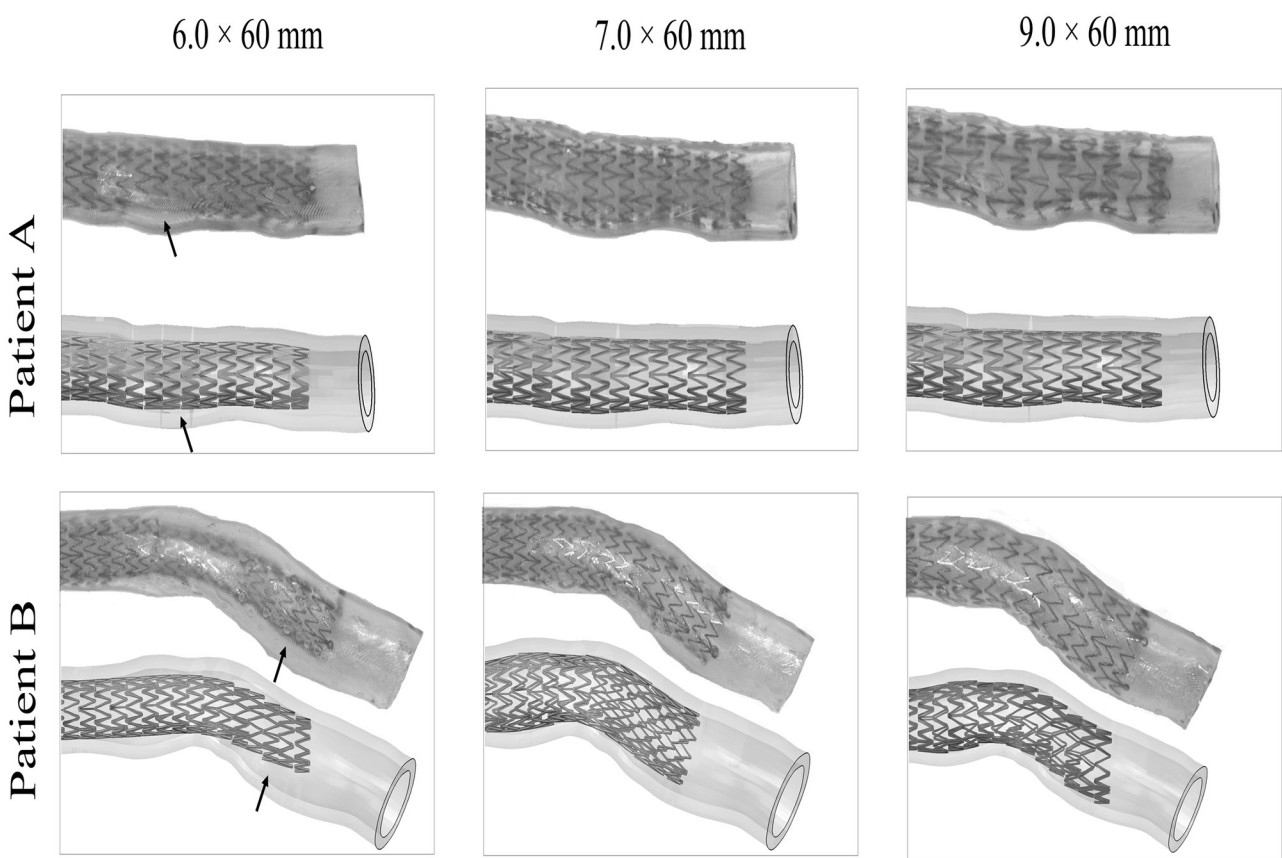

**Fig 13. Validation of stent deployment in the 3D-printed patient-specific vessels: comparison of the deployment for different stent diameter sizes in scenarios A and B.**

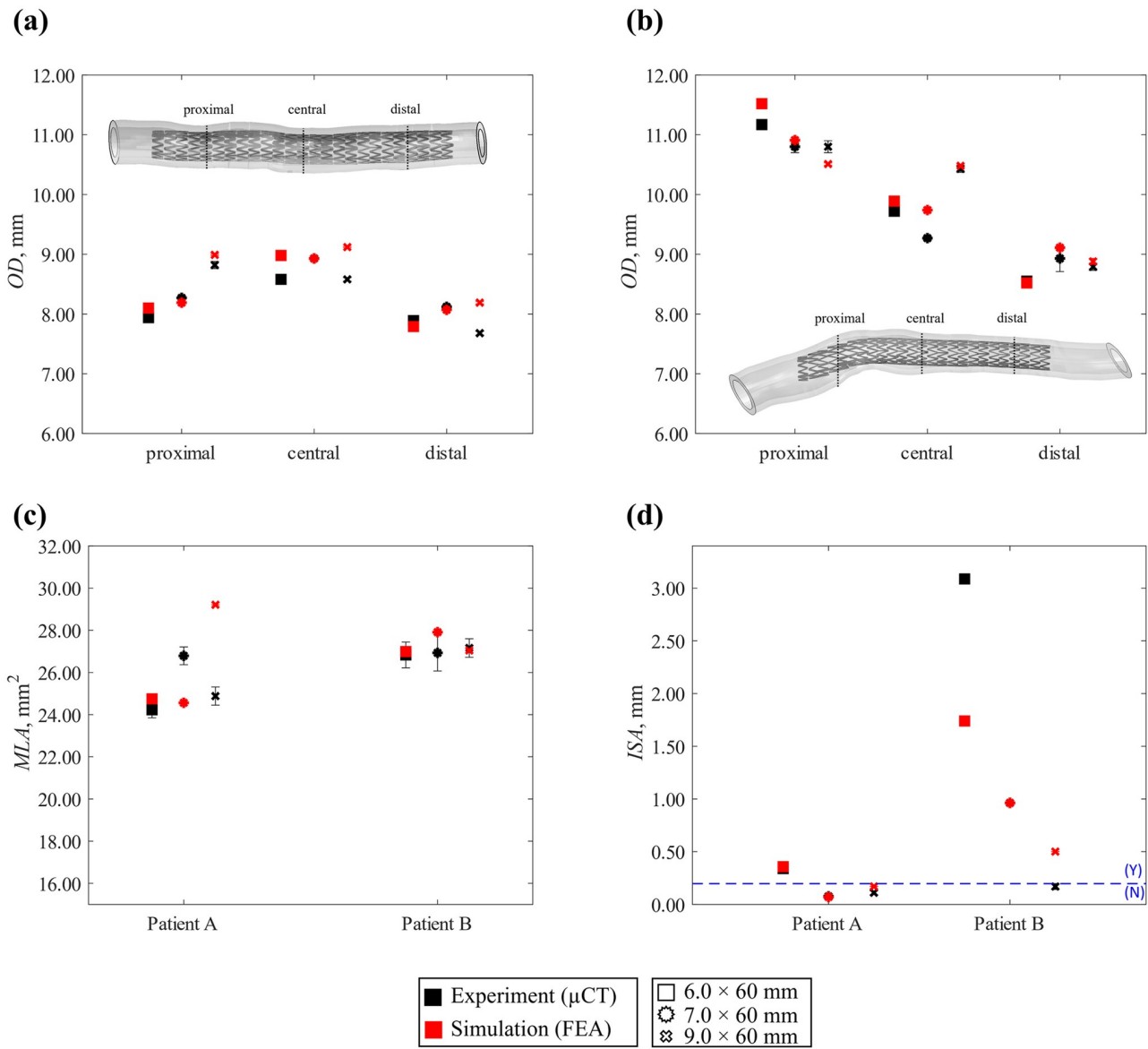

**Fig 14.** Results of in vitro (black) and in silico (red) quantitative indicators: outer diameter measured in proximal, central, and distal position for patient A (a) and B (b); minimal lumen area **(c)**; incomplete stent apposition **(d)**.

prediction of vessel diameter following deployment was least sensitive to numerical parameters. Further detail and discussion on aspects of calculation verification and validation are provided in Sections 5.2 and 5.3.

## 5.2 Calculation verification activities

The temporal and spatial discretization of numerical models of Ni-Ti stents has not been extensively investigated or reported in the literature, with only a limited number of studies considering such details [21, 29, 30]. In this study, a detailed investigation of calculation verification activity was conducted, whereby the effects of mesh density (*MD*), element integration (*EI*) and mass scaling (*TTI*) were addressed for each COU.

**Table 14. Measurement of the error among experimental and computational outcomes.**

| | Validation of Deployment: *in vitro* Patient-specific Resin Vessels | | | | | |
| --- | --- | --- | --- | --- | --- | --- |
| | Patient A | | | Patient B | | |
| Stent Size | 6.0 × 60 mm | 7.0 × 60 mm | 9.0 × 60 mm | 6.0 × 60 mm | 7.0 × 60 mm | 9.0 × 60 mm |
| **Outer Diameter (*OD*)** | | | | | | |
| $\varepsilon_{OD\text{-}proximal}$, % | +2.07% | -1.02% | +1.92% | +3.18% | +1.05% | -2.67% |
| $\varepsilon_{OD\text{-}central}$, % | +4.68% | +0.01% | +6.35% | +1.70% | +4.99% | +0.42% |
| $\varepsilon_{OD\text{-}distal}$, % | -1.28% | -0.63% | +6.53% | -0.45% | +1.97% | +0.95% |
| **Minimum Lumen Area (*MLA*)** | | | | | | |
| $\varepsilon_{MLA}$, % | +2.14% | -8.32% | +17.41% | +0.62% | +3.66% | -0.44% |
| **Incomplete Stent Apposition (*ISA*)** | | | | | | |
| $ISA_{\mu CT}$, mm | 0.34 (Y) | 0.08 (N) | 0.11 (N) | 3.09 (Y) | 0.96 (Y) | 0.17 (N) |
| $ISA_{FEA}$, mm | 0.37 (Y) | 0.07 (N) | 0.17 (N) | 1.74 (Y) | 0.96 (Y) | 0.50 (Y) |

In terms of spatial discretization, it was found that element density and formulation had a significant influence on the predictions of QoIs across each COU. It was found that an element density of 4 × 4 elements across the strut provided optimal results for both COU-1 and COU-2, since it gave a reduction of up to 82% of computation time, while global and local QoIs were predicted within 2% of the solution from the 8 × 8 grid. While this minimum density of 4 × 4 continuum elements has previously been suggested [17, 19, 27], it was observed that the results for fatigue life estimate (COU-3) were more sensitive to the choice of element size, with a 6.5-fold difference in *FSF* of the 4 × 4 elements compared to the 8 × 8 refinement. Together, these results indicate that at least 4 × 4 mesh across the strut width and thickness should be used for the prediction of radial compression/deployment. In contrast, smaller elements with at least 6 × 6 density or finer are suggested for fatigue life estimation, where a reliable prediction of local strain and stress is necessary.

Considering element type, the analysis focused on linear hexahedral elements, which are widely used [14–16, 27], and addressed full and reduced integration formulations. Within the reduced formulation, three different options for the evaluation of the kinematic split (namely *average*, *orthogonal* and *centroid*) were considered, and analyses were run with the 4 × 4-element grid. The use of the centroid kinematic split (Red-C) resulted in the highest deviations in both COU-1 and COU-2, with +49.6% deviation of force at maximum crimp (COU-1) and a deviation of -13.4% of diameter gain (COU-2), compared to default reduced integration (Red-A). For COU-3, the difference was limited to Red-O options which resulted in a 0.5-fold prediction of the *FSF*. Here, the default reduced integration option (Red-A) is recommended as it provided the most appropriate predictions across all the COUs. Additionally, the enhanced hourglass option, which has often been used in the literature (31,52) to counteract numerical artefacts, provided a better description of the radial behaviour, although it resulted in a 3.2-fold rise in computational time when compared to default formulation.

In terms of temporal resolution, explicit solution schemes are most commonly used when modelling Ni-Ti stents, with mass scaling approaches widely implemented to increase the required time increment, enabling more efficient computation [28, 29]. With this approach, the most common verification activity monitors the ratio of kinetic and internal energies to ensure that these are below 5% [70]. In this study, a systematic investigation of the target time increment *(TTI)* selected during the mass scaling procedure was carried out on the 4 × 4-element grid, with model results across all COUs generally being highly sensitive to this parameter. It was found that *TTI* had a substantial impact on the prediction of diameter gain in the

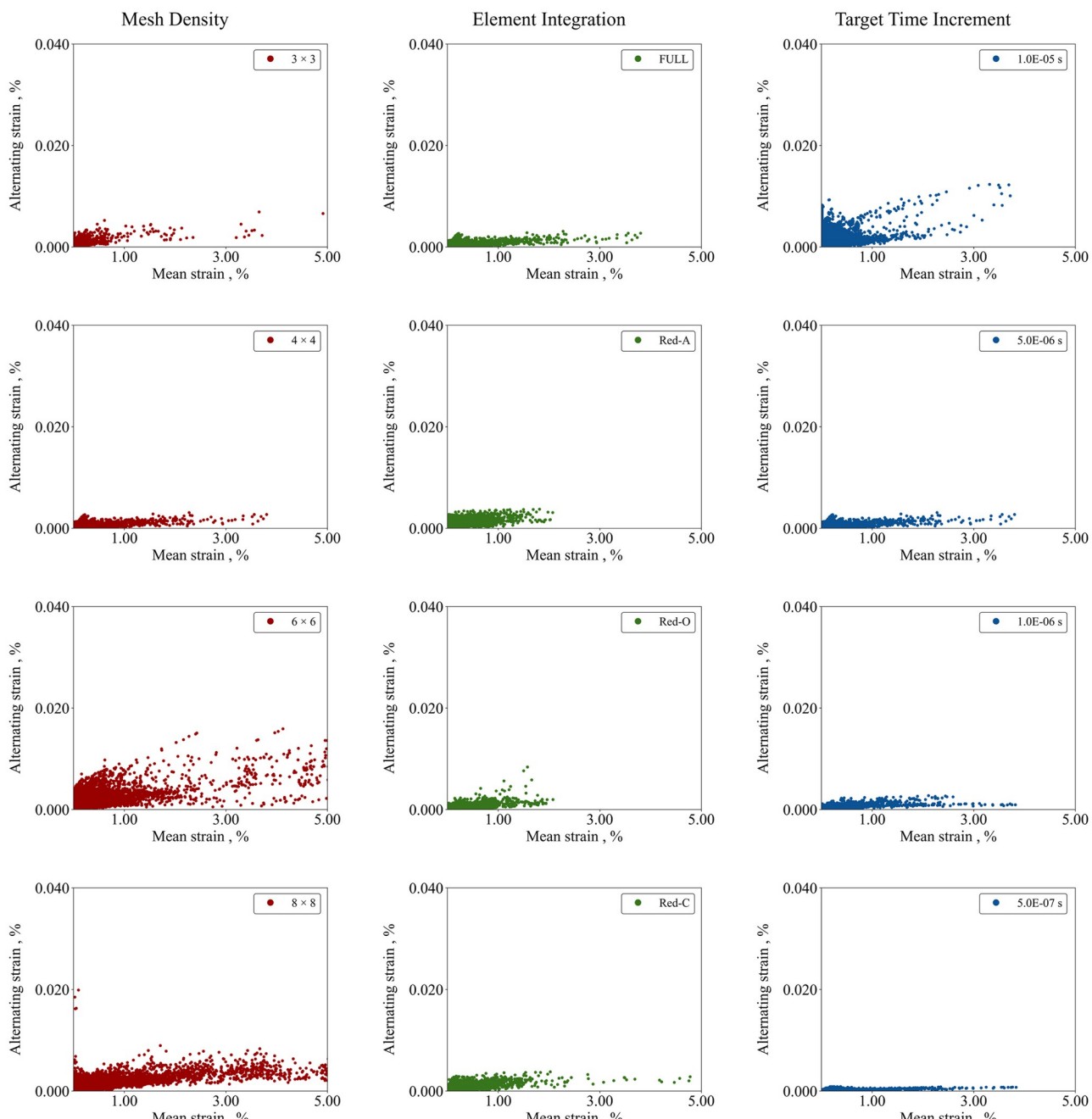

**Fig 15. Calculation verification applied to fatigue-life simulation accounting for the effects of mesh density, element integration and target time increment.**

deployment simulation (COU-2), with up to 30% difference between *TTI* = 1.0E-05 s and smaller values. Furthermore, the stable time increment with *TTI* = 1.0E-5 s under-predicted the *FSF* (COU-3) by over ten-fold compared to the smaller *TTI*. On the other hand, smaller deviations were found for radial behaviour (COU-1) with the larger TTI being within 10% of the reference value from a dynamic implicit solution when predictions of global radial force

**Table 15. Results of the calculation verification activities applied to fatigue analysis.** The fatigue safety factor was normalized according to the reference case, here reported in the table with the symbol $\overline{F}$.

| | | | | | Verification COU-3 | | | | | | |
|---|---|---|---|---|---|---|---|---|---|---|---|
| *MD* | $\varepsilon_{\mathrm{m}}$, % 97.5$^{\mathrm{th}}$ | $\varepsilon_{\mathrm{a}}$, % max | *FSF norm.* | *EI* | $\varepsilon_{\mathrm{m}}$, % 97.5$^{\mathrm{th}}$ | $\varepsilon_{\mathrm{a}}$, % max | *FSF norm.* | *TTI* | $\varepsilon_{\mathrm{m}}$, % 97.5$^{\mathrm{th}}$ | $\varepsilon_{\mathrm{a}}$, % max | *FSF norm.* |
| **3 × 3** | 0.71 | 0.008 | 2.51 $\overline{F}$ | **Full** | 0.95 | 0.003 | 1.22 $\overline{F}$ | **1E-05 s** | 0.88 | 0.012 | 0.07 $\overline{F}$ |
| **4 × 4** | 0.95 | 0.003 | 6.45 $\overline{F}$ | **Red-A** | 0.83 | 0.004 | $\overline{F}$ | **5E-06 s** | 0.95 | 0.003 | 0.28 $\overline{F}$ |
| **6 × 6** | 1.05 | 0.016 | 1.25 $\overline{F}$ | **Red-O** | 0.81 | 0.008 | 0.45 $\overline{F}$ | **1E-06 s** | 1.01 | 0.003 | 0.32 $\overline{F}$ |
| **8 × 8** | 0.90 | 0.020 | $\overline{F}$ | **Red-C** | 0.94 | 0.004 | 0.99 $\overline{F}$ | **5E-07 s** | 1.02 | 0.001 | $\overline{F}$ |

and local prediction of stress at maximum crimping were considered. Importantly, this study demonstrated that substantial deviations in predicted QoIs were observed across different models, even though the ratios of kinetic to internal energies were well below 5% for most of the analysis. This implies that a careful sensitivity analysis for *TTI* is required, irrespective of whether the ratios of kinetic and internal energies are below 5%. It is also important to note that the *TTIs* used here are only relevant to the specific model parameters also selected here and *TTIs* are not broadly applicable to other models. The appropriate selection of mass scaling parameters and *TTIs* is uniquely dependent on the specific parameters of an individual model (e.g., step time, element size, loading amplitudes etc).

## 5.3 Validation activities

**5.3.1 Model input sensitivity.** Validation of the model inputs was carried out to determine the sensitivity of the model to stent geometry and nickel-titanium material parameters for COU-1 and COU-2. The rationale behind this activity was motivated by the fact that model geometries and material-level parameters for nickel-titanium devices are not always widely available.

Considering the geometric features, the dimensions of Ni-Ti stents are often acquired with different imaging techniques [14, 15, 19, 27, 59], which are subject to some degree of uncertainty during measurements. Even when the geometry is known (e.g., CAD is provided by the stent manufacturer), it typically represents the laser-cut configuration that could be altered during further manufacturing processes. For instance, electropolishing can results in differences in overall final outer diameters [71], with removal up to 21 μm possible during this step [60]. Our results showed that a 10% variation in stent thickness altered the prediction of the radial *COF* by up to 16.1%, while it altered the predicted maximum stress by only 5.5%. On the other hand, variation of the strut width had a greater influence on predicted quantities, with deviations up to 25.0% for *COF* while the deviation on the prediction of the maximum stress were in line (+3.5%). Even though the influence of geometric dimensions is largely consistent with their relative contribution to the second moment of area, these results do indicate that the correct geometry is required to correctly predict global responses.

In terms of the Ni-Ti parameters, material-level data of devices is rarely available, and some studies have been forced to calibrate the material parameters to the radial response of the device [13, 40, 72], with certain challenges in this activity given the large number of parameters to be determined. It was generally found in this study that model predictions under radial compression were not overly sensitivity to the nickel-titanium parameter description. Only a few parameters, namely $E_{\mathrm{A}}$ and $\sigma_{\mathrm{L}}^{\mathrm{S}}$, had an influence on the predicted global and local QoIs, although the differences were not larger than 4.9% in force and 9.5% in stress. These results

indicate a certain amount of sensitivity to material inputs, where the behaviour of different regions of the radial force curve depends on specific material parameters. Although, this analysis shows that a calibration approach based on radial compression tests appears suitable to tune the main parameters that influence the device behaviour in crimping and deployment. Nevertheless, when considering more extreme deformation, other parameters such as the martensite modulus $E_M$, the stress at the start of unloading $\sigma_L^S$, or the maximum transformation strain in tension $\varepsilon_L$ might require an appropriate calibration through a different mechanical test.

**5.3.2 *In vitro* comparison of stent deployment into mock vessels.** Several computational models have been employed to predict the deployment of Ni-Ti devices, however, only a limited number of studies have considered *in vitro* approaches for the validation of this simulation [21, 22]. For peripheral stents, whose main function is to restore patency by expanding the inner lumen of the vessel, deployment simulations have been used to predict parameters such as the lumen gain [16, 28, 42], though an analysis of the accuracy of such predictions has not be reported. Recently, a quantitative validation approach was proposed for balloon-expandable coronary stents by Berti *et al.* [38], who measured the outer diameter post-deployment in a mock vessel comparator. In our study, we have employed a similar strategy whereby the predicted deployment configuration was directly compared to straight and 3D-printed mock vessel comparators. For the patient-specific vessels, several testing conditions were taken into consideration, which included different geometries and different stent nominal diameter sizes, thereby increasing the credibility of the model since each scenario represented an independent validation point [8]. Each material for the mock vessel, namely silicone for the straight vessel and 3D-printed resin for patient-specific geometry was characterised through tensile tests either on a sample cut from the vessel (silicone) or from dog-bone specimens that were 3D-printed with the same settings as the vessels (e.g., printing angle, curing temperature and time). Furthermore, the numerical model parameters of these simulations were those identified to be optimal based on the previous verification activities.

Straight vessel scenario showed that measurements of outer diameter (*OD*) taken on *in vitro* comparators have inherent relative uncertainty of approx. 0.8%. Based on this, it was found that the predicted deployment in straight silicone mock vessel closely agreed with the *in vitro* comparators, with the prediction of outer diameter (*OD*) following deployment resulting in an average error of only 1.04%.

Deployment into patient-specific 3D-printed vessels was used to assess *OD* prediction, as previously performed [38], as well as minimum lumen area (*MLA*) and incomplete stent apposition (*ISA*) [37], which are highly relevant from a clinical point of view. Again, it was found that the measurement of *OD* was generally accurate across all mock vessels, irrespective of the acquisition method (measurement from optical images vs. measurements on cross-sectional images acquired with micro-CT), with only an average error of 1.65% across the scenarios (A and B) and the different stent sizes considered. It was also found that the model could accurately predict the *MLA*, with only an average error of 2.08%. In predicting malapposition to the vessel wall, it was determined that the model could correctly predict the locations in which *ISA* occurred across all models. Although, the precise magnitude of the *ISA* had an average error of 33.7%, with these errors likely arising due to uncertainties related to stent positioning in the vessel (e.g., use of centreline vs. guidewire for the *in silico* deployment) and the very small resolution of these measurements. These results suggest that in the context of stent deployment, a model validated with *in vitro* idealised (i.e., cylindrical, and straight) vessels will provide an accurate prediction of diameter gain and lumen area. However, the prediction of local deployment features, such as malapposition, is more challenging and requires sophisticated models and experimental comparators to ensure credibility.

## 5.4 Limitations

Among the limitations of the study, the following aspects are worth consideration. A reduced unit of the stent (and vessel) and symmetric boundary conditions around the circumferential axis were applied for calculation verification and sensitivity analysis. Despite being a simplification of the more complex response of a full stent, the radial behaviour appropriately scaled (e.g., radial force normalization) was deemed to be representative of experimental outcomes while providing efficient computational times. In the calculation verification, predictions could not be evaluated against a closed-form analytical solution so, instead, the QoIs from an explicit solution scheme was compared to the more highly resolved versions of an implicit analysis. While this provides an indication of QoIs converging towards a solution (where error becomes progressively minimised) for both stable time increment and mesh density, the analysis around element type does not yield a converged solution but instead shows differences in predictions for different element formulations. Also, for verification activities, the results hereby presented were obtained with Abaqus, which could provide different outcomes when different software is chosen. While the methodology is applicable to different stent designs, the results here reported (e.g., sensitivity analysis on model inputs) are representative of the specific device accounted.

## 6. Conclusion and recommendations

This work focused on a self-expanding nickel-titanium stent and developed a finite element model to address the credibility steps required for contexts of use (COUs) commonly reported in the literature, namely radial compression test (COU-1), stent deployment in a vessel (COU-2) and fatigue life predictions (COU-3). As per the VV-40 standard [11], a series of verification and validation activities were performed, and their relevance was assessed over the COUs and their quantities of interests (QoIs) and several recommendations are have been made.

In terms of calculation verification, it is recommended to:

- Use at least $4 \times 4$ element density across the strut width and thickness for prediction of radial compression/deployment; while a $6 \times 6$ element density or finer are more appropriate for fatigue life estimation where a reliable prediction strain/stress is necessary.

- Conduct a careful sensitivity analysis on the effects of mass scaling options on the QoIs, irrespective of whether the ratios of kinetic and internal energies are below 5%.

- Use reduced integration over full integration continuum hexahedral elements to avoid overly stiff behaviour in bending, with enhanced hourglass control options providing additional benefits, although at increased computational cost.

Considering model inputs and model validation,

- The sensitivity analysis on geometrical and material inputs highlighted the importance of correctly capturing the geometric dimensions of the stent, in particular the strut width, while model results were less sensitive to Ni-Ti material parameters, suggesting that device-level calibration of material behaviour are reasonable.

- The validation of stent deployment through the use of idealised vessel geometries offers a simple and accurate validation method when predicting diameter gain, and lumen area, provided that the material of the vessel is appropriately characterized and modelled. However, predictions of local stent-to-lumen interactions require more sophisticated validation approaches.

## Supporting information

**S1 Appendix. Contains all the supporting tables and figures.**
(DOCX)

## Acknowledgments

The authors wish to thank Dr. Monika Colombo and Prof. Claudio Chiastra for providing the patient-specific vessel geometries. The authors wish to acknowledge the Irish Centre for High-End Computing (ICHEC) for the provision of computational facilities.

## Author Contributions

**Conceptualization:** Martina Bernini, William Ronan, Ted J. Vaughan.

**Data curation:** Martina Bernini.

**Formal analysis:** Martina Bernini.

**Funding acquisition:** Ted J. Vaughan.

**Investigation:** Martina Bernini.

**Methodology:** Martina Bernini, Rudolf Hellmuth, Craig Dunlop, Ted J. Vaughan.

**Project administration:** Ted J. Vaughan.

**Resources:** Rudolf Hellmuth, Craig Dunlop.

**Supervision:** William Ronan, Ted J. Vaughan.

**Writing – original draft:** Martina Bernini, Ted J. Vaughan.

**Writing – review & editing:** Martina Bernini, Rudolf Hellmuth, William Ronan, Ted J. Vaughan.

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
