## [Decision Letter · Decision Letter 0]

27 Jan 2023

PONE-D-23-00655Recommendations for finite element modelling of nickel-titanium stents – verification and validation activitiesPLOS ONE

Dear Dr. Bernini,

Thank you for submitting your manuscript to PLOS ONE. After careful consideration, we feel that it has merit but does not fully meet PLOS ONE’s publication criteria as it currently stands. Therefore, we invite you to submit a revised version of the manuscript that addresses the points raised during the review process.

Please, address all the comments made by both reviewers.

We look forward to receiving your revised manuscript.

Kind regards,

Antonio Riveiro Rodríguez, PhD

Academic Editor

PLOS ONE

Journal Requirements:

2. Please ensure that you have specified (1) whether consent was informed and (2) what type you obtained (for instance, written or verbal, and if verbal, how it was documented and witnessed). If your study included minors, state whether you obtained consent from parents or guardians. If the need for consent was waived by the ethics committee, please include this information.

"TJV reports financial support was provided by EU Framework Programme for Research and Innovation Marie Sklodowska-Curie Actions. This study is part of the BioImplant ITN project, which has received funding from the European Union’s Horizon 2020 research and innovation programme under grant agreement No 813869."

"This study is part of the BioImplant ITN project, which has received funding from the European Union’s Horizon 2020 research and innovation programme under grant agreement No 813869. The authors wish to thank Dr. Monika Colombo and Prof. Claudio Chiastra for providing the patient-specific vessel geometries. This publication reflects only the author’s view, and the REA is not responsible for any use that may be made of the information it contains. The authors wish to acknowledge the Irish Centre for High-End Computing (ICHEC) for the provision of computational facilities."

"TJV reports financial support was provided by EU Framework Programme for Research and Innovation Marie Sklodowska-Curie Actions. This study is part of the BioImplant ITN project, which has received funding from the European Union’s Horizon 2020 research and innovation programme under grant agreement No 813869."

Reviewers' comments:

Reviewer's Responses to Questions

**Comments to the Author**

1. Is the manuscript technically sound, and do the data support the conclusions?

Reviewer #1: Yes

Reviewer #2: Yes

2. Has the statistical analysis been performed appropriately and rigorously? 

Reviewer #1: Yes

Reviewer #2: N/A

3. Have the authors made all data underlying the findings in their manuscript fully available?

Reviewer #1: No

Reviewer #2: Yes

4. Is the manuscript presented in an intelligible fashion and written in standard English?

Reviewer #1: Yes

Reviewer #2: Yes

5. Review Comments to the Author

Reviewer #1: Explanation to question 3: In order to make all data available I consider that the CAD geometry of the stent should be shared or uploaded in a public repository.

Minor points:

There are two references to Figure 5d on lines 287 and 385. However the attatech Figure 5 has only labels a) b) and c).

Table 12: Lumen Gain should be defined in the text. In reference 44 the Lumen Gain is defined as a dimensionless quantity.

I think that the version code of the Commercial Software Package used for the simulations should be mentioned.

Suggestion for COU-1: a verification activity comparing the results of the functional unit (axisymmetric boundary conditions) with the results obtained with the complete geometry of the stent.

In Table 7 and Table 8 Chronic Outward Force is tabulated but in Figures 6 and 7 Radial Force along the test is plotted. Is there any way that the values of the Chronic Outward Force appear in the Figures?

Reviewer #2: In the paper, the finite element modeling of Nitinol stents were studied in verification and validation. It's an interesting work and useful for those interted in developing stents. I have some question on the FEM: 1) how to justify one unit cell is good enough to represent the whole stent? 2) the stent is not axisymmetric but can be considered as cyclic symmetric (rotational). 3) the fig1c is not corresponding to the rectangular region of fig1a.

6. PLOS authors have the option to publish the peer review history of their article (what does this mean?). If published, this will include your full peer review and any attached files.

Reviewer #1: No

Reviewer #2: No

---

## [Author Response · Author response to Decision Letter 0]

18 Feb 2023

All reviewers' and editors' comments have been addressed in the uploaded 'Response_to_Reviewers' file.

---

## [Decision Letter · Decision Letter 1]

13 Mar 2023

Recommendations for finite element modelling of nickel-titanium stents – verification and validation activities

PONE-D-23-00655R1

Dear Dr. Bernini,

We’re pleased to inform you that your manuscript has been judged scientifically suitable for publication and will be formally accepted for publication once it meets all outstanding technical requirements.

Kind regards,

Antonio Riveiro Rodríguez, PhD

Academic Editor

PLOS ONE

Reviewers' comments:

Reviewer's Responses to Questions

**Comments to the Author**

1. If the authors have adequately addressed your comments raised in a previous round of review and you feel that this manuscript is now acceptable for publication, you may indicate that here to bypass the “Comments to the Author” section, enter your conflict of interest statement in the “Confidential to Editor” section, and submit your "Accept" recommendation.

Reviewer #1: All comments have been addressed

Reviewer #2: All comments have been addressed

2. Is the manuscript technically sound, and do the data support the conclusions?

Reviewer #1: (No Response)

Reviewer #2: Yes

3. Has the statistical analysis been performed appropriately and rigorously? 

Reviewer #1: (No Response)

Reviewer #2: N/A

4. Have the authors made all data underlying the findings in their manuscript fully available?

Reviewer #1: (No Response)

Reviewer #2: Yes

5. Is the manuscript presented in an intelligible fashion and written in standard English?

Reviewer #1: (No Response)

Reviewer #2: Yes

6. Review Comments to the Author

Reviewer #1: All issues and comments from minor revisions have been satisfactorily addressed. In my opinion the supporting information with the link to the CAD model and the verification of the unit cell domain is a very suitable choice to provide relevant data without compromising document readability.

Reviewer #2: I still have a question regarding the cyclic symmetry of unit cell. If the authors use ABAQUS software, a method to handle this can be found by defining master and slave surfaces in Interface module. For example, the work by Praveen Kumar G, et al. 2016. Feasibility of using bulk metallic glass for

self-expandable stent applications. J Biomed Mater Res Part B 2016. The benefit of doing this is that results can be post-processed to form a complete stent rather than a unit cell. Besides, it can handle cells with more general design.

7. PLOS authors have the option to publish the peer review history of their article (what does this mean?). If published, this will include your full peer review and any attached files.

Reviewer #1: No

Reviewer #2: **Yes: **F Cui

---

## [Editor Report · Acceptance letter]

17 Mar 2023

PONE-D-23-00655R1 

Recommendations for finite element modelling of nickel-titanium stents – verification and validation activities 

Dear Dr. Vaughan:

I'm pleased to inform you that your manuscript has been deemed suitable for publication in PLOS ONE. Congratulations! Your manuscript is now with our production department. 

Kind regards, 

on behalf of

Dr. Antonio Riveiro Rodríguez 

Academic Editor

PLOS ONE